# Is Developmental Dyslexia Due to a Visual and Not a Phonological Impairment?

**DOI:** 10.3390/brainsci11101313

**Published:** 2021-10-02

**Authors:** Reinhard Werth

**Affiliations:** Institute for Social Pediatrics and Adolescent Medicine, University of Munich, Haydnstrasse 5, D-80336 Munich, Germany; r.werth@lrz.uni-muenchen.de; Tel.: +49-173-3550-232; Fax: +49-308-337-940

**Keywords:** dyslexia, reading impairment, visual system, magnocells, visual word form area, simultaneous recognition

## Abstract

It is a widely held belief that developmental dyslexia (DD) is a phonological disorder in which readers have difficulty associating graphemes with their corresponding phonemes. In contrast, the magnocellular theory of dyslexia assumes that DD is a visual disorder caused by dysfunctional magnocellular neural pathways. The review explores arguments for and against these theories. Recent results have shown that DD is caused by (1) a reduced ability to simultaneously recognize sequences of letters that make up words, (2) longer fixation times required to simultaneously recognize strings of letters, and (3) amplitudes of saccades that do not match the number of simultaneously recognized letters. It was shown that pseudowords that could not be recognized simultaneously were recognized almost without errors when the fixation time was extended. However, there is an individual maximum number of letters that each reader with DD can recognize simultaneously. Findings on the neurobiological basis of temporal summation have shown that a necessary prolongation of fixation times is due to impaired processing mechanisms of the visual system, presumably involving magnocells and parvocells. An area in the mid-fusiform gyrus also appears to play a significant role in the ability to simultaneously recognize words and pseudowords. The results also contradict the assumption that DD is due to a lack of eye movement control. The present research does not support the assumption that DD is caused by a phonological disorder but shows that DD is due to a visual processing dysfunction.

## 1. Introduction

Reading disorders, like all cognitive disorders, are caused by a dysfunctional neural network. The Diagnostic and Statistical Manual of Mental Disorders (DSM5) [1] defines developmental dyslexia (DD) primarily by the exclusion of deficits. For the diagnosis of DD, the DSM5 requires not only that reading problems are not caused by intellectual disabilities, poor visual or auditory acuity, psychological adversity, or inadequate educational instruction, but also that the reading problems are not due to psychiatric and neurologic disorders. According to this definition, DD is an independent disorder that is not the consequence of another disorder, such as visual impairment.

A widely held opinion is that DD is a phonological disorder that presents as an impaired ability to associate letter sequences with correct sound sequences [2,3,4,5,6,7,8,9,10,11,12,13,14,15,16]. Phonological awareness enables children to learn grapheme-phoneme correspondence and to use it when reading or spelling. Phonological awareness does not designate a single ability, but comprises different abilities that are assumed to promote reading skills [6], such as decomposing words into syllables and sounds [2,5,9,11,17,18,19], identifying phonemes in words [3,7,10], naming letters, objects, numbers and colors [8,18] and rhyming [4]. An impairment of these abilities may coexist with DD, but no causal relationship between them and DD has been demonstrated. Many correlations between DD and other impairments have been reported without demonstrating a causal relationship [20,21,22,23,24,25,26,27,28,29,30,31,32,33,34,35,36,37,38,39,40,41,42,43]. These correlations do not explain whether DD is the consequence of other performance impairments or contribute to revealing the neurobiological causes of DD. Heim et al. (2008) [44] distinguished between different kinds of DD based on other impairments that coexist with DD. However, the simultaneous existence of DD and various other impairments does not mean that these impairments cause different kinds of reading problems. Reading problems may be causally independent of other coexisting impairments as long as no causal relationship is demonstrated. To demonstrate the existence of a causal relationship between a performance deficit and DD, the necessary and sufficient conditions for a normal reading performance must be identified, and it must be shown that DD is present whenever at least one necessary condition, or all sufficient conditions for normal reading are missing. These logical relationships between DD and other performances can only be investigated if they are based on a precise concept of causation [45,46,47,48] (Appendix A).

The magnocellular theory of DD assumes that DD is a disorder of visual stimulus processing that is due to the reduced function of the magnocellular visual pathway [49,50,51,52,53,54,55]. Recent findings [56,57,58,59,60] support the view that DD is caused by a visual processing disorder and not a reduced ability to associate a sequence of letters with a sequence of sounds. DD was caused by a visual impairment in all 356 children who participated in these studies and received a diagnosis of DD, although visual acuity was normal and there was no motor eye movement disorder. Instead the visual impairments consisted of an increased required fixation time, premature onset of saccades, poor ability to recognize a sequence of letters simultaneously, and saccades whose amplitudes did not match the ability to recognize a string of letters of a given length simultaneously. When visual impairments were compensated by a computer-controlled altered reading strategy, reading performance improved immediately without reading training [59]. Reading training lasted less than 30 min in other studies [56,57,58,60]. Such short training cannot overcome neural dysfunction, leading to a diminished ability to associate a sequence of letters with a sequence of sounds. These results contradict the hypothesis that DD is due to a phonological disorder and point to impaired visual processing, possibly caused by a functional impairment of the magnocell- and the parvocell-system.

Some children have problems storing some familiar grapheme-phoneme connections in memory. German-speaking children typically interchange the letters “b” with “d”, “p” with “q”, “m” with “n”, and “u” with “v”. The children can see the letters clearly but do not know which phoneme is associated with the letter. A text that does not contain these critical letters can be read normally. Children who can see words clearly without knowing the phonemes belonging to the sequence of letters that make up the word are by no means typical for children with DD. None of the 356 children in our studies [56,57,58,59,60] suffered from such a phonological impairment but most of them still had a severe DD.

The aim of the review is to examine whether there is evidence to support the hypothesis that DD is caused by a phonological disorder, an impairment of the magnocellular stream, or an impairment of the visual system and the visual word form area (VWFA). The review also examines whether the studies that claim to shed light on the causal relationship between reading performance and neurobiological processes satisfy the conditions that must be met to claim such a causal relationship.

To this end, several thousand studies about the anatomy, physiology, and neuropsychology of the visual system in humans, visual psychophysics and dyslexia, which were available in the Max-Planck-Institute for Psychiatry, the Bavarian State Library, the Library of the Medical Faculty of the University of Munich, Pubmed, Science Direct, Psycnet or other internet-based databases, and which the author collected over about 40 years up to the year 2021, were checked to determine whether they were relevant to the questions posed in the present review. In total, 286 of the most relevant studies were included.

## 2. Origin of the Magnocellular, Parvocellular, and Koniocellular Pathways in the Retina

Neural dysfunction of the visual system that causes DD may involve the magnocellular-, parvocellular-, and koniocellular systems and a disturbed interaction between these systems. They originate in the retina and continue in the lateral geniculate body (LGN) and primary and secondary visual cortex.

When reading, we must move the word or word segment to be read into the foveal area of the retina because a sufficiently high visual acuity only exists here. The fovea and parafoveal region play a fundamental role in reading. Although the fovea accounts for only 1% of the retinal area, 50% of the input to area V1 of the visual cortex comes from the fovea [61]. Retinal ganglion cells are divided into macrocells (M-cells), which are anatomically also called “parasol cells”, parvocells (P-cells), which are anatomically called “midget cells”, and koniocells (K-cells). At least 80% of the ganglion cells in the human fovea are midget parvocells (P-cells) [62,63,64,65]. The remaining 10–20% of ganglion cells are parasol magnocells (M-cells). M-cells have larger cell bodies, greater dendritic arborization, and wider receptive fields than P-cells, which have rather small receptive fields. M-cell axons conduct information faster than P-cell axons. P-cells, in contrast to M-cells, P-cells can convey color information, have high visual acuity, and provide an accurate analysis of visual stimuli. M-cells convey information about fast-moving stimuli in the visual field but contribute little to visual acuity. M-cells are relatively more sensitive to high temporal frequencies and show higher activity on temporal contrast than P-cells. P-cells have higher spatial frequency tuning than M-cells, which have a higher ability to detect flicker and motion in visual space [62,64,65,66]. There is also an eccentricity-dependent increase in the ratio of M- (parasol) to P- (midget) retinal ganglion cells. Foveal P- (midget) retinal ganglion cells receive input from a single cone photoreceptor via a midget bipolar cell, which is the basis for the high spatial acuity of foveal vision. In contrast, central P- (midget) retinal ganglion cells outside the fovea and peripheral P- (midget) retinal ganglion cells receive input from 2–6 cones and 10–30 cones, respectively [67,68,69,70,71,72]. At 10 degrees eccentricity, temporal contrast sensitivity is due to the function of M- (parasol) retinal ganglion cells [73]. Magnocells project to the two most ventral magnocell layers of the lateral geniculate nucleus (LGN), whereas midget parvocells send fibers to the other four layers [74,75]. Besides M-cells and P-cells, the retina also contains so-called “koniocells” (K-cells) that transmit visual information to thin koniocellular layers of the LGN and further on to area V1 of the visual cortex and to the extrastriate middle temporal area (MT/V5) [76,77,78].

## 3. Neural Wiring in the Visual Cortex

In the monkey, the afferents of M-magnocells and P-parvovells from the LGN terminate in different sublayers of layer 4 of the visual cortex (area V1) and remain strictly segregated within this layer [79,80,81]. The fibers of the koniocellular (K) pathway, which have been described in the prosimian bushbaby (*Galago*) and simian primates (macaques), also project from the LGN to area V1, where they terminate in color-selective blobs [82,83]. Monkey afferents from the M- and P-cell layers of the LGN also project to visual areas V2 on the posterior bank of the lunate sulcus, area V3 [84,85,86], and area V4 [87,88]. Areas V2 and V3 project to V4 and to the middle temporal area (MT/V5), an area in the anterior occipital sulcus [89,90,91,92]. MT/V5 predominantly receives fibers from the magnocellular layers of the LGN whereas parvocellular fibers seem to have little influence on MT/V5 activity, and project from area V1 to area V4 [93,94]. From a region in area V1, which represents the center of gaze (fovea), fibers reach the posterior bank of the lunate sulcus (V2), inferior occipital sulcus, color sensitive area V4 in the anterior bank of the lunate sulcus, the posterior bank of the superior temporal sulcus, posterior bank of the inferior occipital sulcus and the posterior bank of the superior temporal sulcus (STS) (Figure 1). Only the foveal representation in V1 sends fibers to V4, which also receives direct input from visual area V2. A region that represents the vertical meridian 2 degrees above the center of gaze projects to the inferior occipital sulcus (V2), the depth of the occipital sulcus (V3), and the posterior bank of the superior temporal sulcus [95,96,97,98,99,100,101,102].

In the monkey, the occipitotemporal pathway from area V1 via areas V2 and V4 into area TEO appears to be the mainstream route. TEO is an area situated between the anterior inferotemporal cortex, and the ventral region of area V4 [103]. A small number of cells project along a bypass route from the foveal representation in Vl to V4 and cells in V2 project to TEO [91,104]. MT/V5 in turn projects to the medial superior temporal area (MST), to the (floor of the) STS and area TEO between the anterior inferotemporal cortex and the ventral region of area V4 [103]. Areas V4, TEO, and the inferotemporal gyrus (TE) on the inferior temporal cortex project to the STS, which projects to TE and the ventral temporal pole (TG) [90,91]. In the monkey, area TEO is topographically and reciprocally connected with areas V2, V3, and V4 and has a sparser connection with areas V3A, V4t, and MT/V5. Thus, TEO receives visual information from V1 via V2, V4, and MT/V5. TEO and TE mainly receive fibers from the representation of the fovea. TEO is also reciprocally connected to the posteromedial superior temporal sulcus, intraparietal sulcus, frontal eye field, and parahippocampal gyrus. TEO sends fibers to the area TE and the lateral bank of the superior temporal sulcus and receives fibers from these areas. TEO connects the occipitotemporal pathway, which mediates object recognition, and sends visual information from V1, V2, V3, V3A and V4 to the anterior inferior temporal area TE [91,105]. Cells in the middle temporal area (in the posterior bank of the STS) are movement and direction selective, but are not selective for shapes, sizes, and contrasts of visual stimuli. They appear to play an important role in remembering the direction of visual motion [106,107,108].

In humans, there is also a foveal representation in areas V1, V2, and V3, and M-cells of the human visual area V1 send fibers to areas V2 and MT/V5, an area on the anterior occipital sulcus [89,109]. Bar et al. (2001) [110] demonstrated that activation of areas V1 and V2, ventral regions of area V4 and area LO (i.e., an area of the lateral occipital cortex that is located ventrally and posteriorly to area MT/V5) depends on whether a picture is seen clearly. In this study, pictures were presented only once or repeatedly to adult subjects. Presentation times were between 26 ms and 221 ms. The subjects were asked to indicate whether they did not recognize the drawings at all, only vaguely, or clearly. Functional magnetic resonance imaging (fMRI) showed that activation of areas V1, V2, ventral regions of area V4, and area LO increased the more clearly the drawings were seen.

There are many more interconnections in the visual system such that the activity of cells in one visual area is not driven solely by input from one area. Felleman and Van Essen (1992) [111] identified 32 areas in the macaque cortex that either contain visually responsive neurons or receive projections from visual areas. Twenty-five of these areas were predominantly or exclusively visual. There were 305 interconnections between visual areas, with 242 of them bidirectional.

## 4. Two Visual Systems

Schneider (1968) [112] proposed a two visual systems theory, in which he distinguished between a where- and a what-system in hamsters. Undercutting the superior colliculi resulted in difficulties in orientation toward an object, but left the ability to visually discriminate between objects intact. Ablation of the visual cortex led to problems distinguishing objects visually but preserved the ability to visually orient to objects. Schneider concluded that the where-system required the superior colliculi while the what-system required the visual cortex. It has been demonstrated that humans can also localize visual stimuli without consciously seeing them [113,114,115]. These visual abilities are probably mediated by the superior colliculi and the praetectum corresponding to Schneider′s where- system. Werth (2007) [116] demonstrated that humans could locate auditory stimuli and exert eye and head movements to a stimulus in the absence of the cortex and white matter of both cerebral hemispheres. The optic nerves and the brainstem were preserved. If an acoustic stimulus of 84 dB was presented 40 times to his left or right ear, the child always turned his head toward the stimulus. When no stimulus was presented in the control trials, no head movement occurred. This performance could only be mediated by the where-system of the brainstem. Ungerleider and Mishkin (1982, 1983) [117,118] also proposed two visual systems: a dorsal stream that includes the striate, prestriate, and inferior parietal cortex, and a ventral stream that connects the striate and prestriate areas with the inferior temporal cortex. The dorsal stream was assumed to mediate the location of objects, whereas the ventral stream was supposed to mediate the identification of objects. Goodale and Milner (1992) [119] endorsed the two visual systems model in which projections from the striate cortex to the inferotemporal cortex (ventral stream) play the key role in the visual identification of objects. According to this hypothesis, the projection from the striate cortex to the posterior parietal region (dorsal stream) mediates visually guided actions directed at objects and mediates the where-information, while the ventral stream is assumed to mediate the what-information [119]. The model was supported by the case of a patient who suffered from bilateral damage to the occipitotemporal visual system (ventral stream), while her occipitoparietal visual system (dorsal stream) remained intact. The patient was able to grasp objects but she was unable to visually discriminate the same objects. fMRI showed activation of areas belonging to the dorsal stream when the patient was grasping objects [120,121,122].

Fridriksson et al. (2016) [123] investigated brain areas involved in speech production. The study was based on MRI data and behavioral testing of 136 post-stroke survivors. The authors distinguished between the dorsal fronto-parietal stream and ventral temporal-frontal stream. The dorsal stream included the pars opercularis and premotor areas, and posterior regions in the supramarginal gyrus. The ventral stream includes the lateral temporal lobe, inferior parietal lobule, uncinate fasciculus, and the inferior frontal lobe. Ventral regions were involved in lexical–semantic analysis of speach, whereas dorsal regions were involved in the phonological–motor processing of speech production.

Choi et al. (2020) [124] proposed a modified model of dorsal stream connectivity in the human brain, in which the angular gyrus is the hub of the where information. In their model, the dorsal stream consists of a projection from the primary visual cortex to area V2, from V2 to area V3, and to MT/V5. There is also a direct connection from V2 to the middle temporal visual area (MT/V5), continuing to the angular gyrus (AG). MT/V5 does not directly project to the superior parietal lobule but only via the angular gyrus. The primary visual cortex projects to V2 and V3. Area V3 sends fibers to MT/V5 in the dorsal stream and to V4 in the ventral stream. In the ventral stream, V4 projects to the inferior temporal gyrus (IT) that projects to AG of the dorsal stream. According to Choi et al. (2020) MT does not directly project to the superior parietal lobule of the dorsal stream (Figure 1).

The dorsal and the ventral streams are not isolated neural systems. The human dorsal and ventral streams are connected by a fiber tract arising from medial bank of the intraparietal sulcus (IPS) connecting the IPS with the fusiform gyrus in the ventral temporal cortex. This connection was believed to mediate sensorimotor integration in visually guided behavior [125]. The angular gyrus, the supramarginal gyrus, and the banks of the intraparietal sulcus (IPS) of the human brain are subdivisions of the inferior parietal lobe (IPL), which is situated at the junction of the temporal, parietal, and occipital lobes. The posterior intraparietal sulcus has stronger connections to the occipital pole and adjacent extrastriate visual areas than the anterior intraparietal sulcus. Two anterior subdivisions of the intraparietal sulcus are connected to the ventral premotor and middle frontal gyrus. The anterior angular gyrus is connected to the cingulate gyrus, frontal gyrus, the bilateral frontal pole, left middle and inferior frontal gyrus, left anterior insula, left posterior middle, and inferior temporal gyrus [126].

The superior longitudinal fasciculus connects the posterior part of the middle and inferior temporal gyri to the angular gyrus and supramarginal gyrus. The vertical occipital fassciculus (VOF) is a major white matter tract connecting the dorsal with the ventral stream in humans. The VOF connects the inferior parietal lobe to the lower temporal and occipital lobe, and a fiber tract connects the inferior temporal gyrus, middle temporal gyrus and fusiform gyrus, and the inferior occipital lobe to the superior parietal lobe [127,130,131].

The ventral projections of the VOF connect the inferior occipital gyrus, inferior occipital sulcus, and posterior transverse collateral sulcus to the posterior mid-fusiform sulcus, lateral regions of the posterior fusiform gyrus, and posterior occipitotemporal sulcus. The dorsal part of the vertical occipital fasciculus projects from the middle occipital gyrus and the lateral occipital sulcus to the transverse occipital sulcus (located at the border between areas V3A and V3B) and the posterior intraparietal sulcus, but does not reach the angular gyrus [132,133].

## 5. Processing of Motion, Contrast, and Patterns in the Dorsal and Ventral Streams

It has been shown that the human area V3A is motion-selective, preferring high motion and high contrast sensitivity in the central visual field, whereas the human V3 is less selective for motion [109]. In humans, there is a rank from high motion selectivity in area MT/V5 to decreasing motion selectivity in areas V3A, V2, and VP (an area corresponding to the ventral area V3 [128]) and the lowest motion selectivity in areas V3 and V1 [109,134].

High-contrast selective cells in humans were found in areas V1 to V4 of the ventral stream, in V3 and V3A, and the intraparietal sulcus in the dorsal stream. Areas V1-Vp and V4 displayed strong contrast dependence at suprathreshold contrast levels, for both faces and objects [135]. There is a growing increase in the level of contrast sensitivity from V2 to V3 to V3A to V4 [136,137]. The highest contrast sensitivity was found for 9 Hz and 6.6 Hz at 2 degrees and 10 degrees eccentricity, respectively. Contrast sensitivity was highest at approximately 8 Hz, increasing between the fovea and the periphery. Similar sensitivity found in the retina exists in the early visual cortex, with increased contrast sensitivity at 20 Hz stimuli in the periphery. V4 appears to be important for the perception of patterns, forms, and colors [138]. Less activation was found in the posterior fusiform gyrus, collateral sulcus, and lateral occipital cortex (LO) (located ventrally and posteriorly to area MT/V5), extending in the posterior inferotemporal sulcus, and into the vicinity of the posterior fusiform gyrus. The lateral occipital cortex and posterior fusiform gyrus showed the highest selectivity for faces, but LO exhibited the same activation for both faces and objects. The posterior fusiform gyrus is anterior and lateral to area V4, extends into the inferior temporal sulcus, and may overlap the fusiform face area [135].

Neurons in area V4 display an increased response to high spatial frequencies. The optimal temporal frequency increased with eccentricity in areas V1 and V2. In areas V3 and V3a, the optimal temporal frequency increased from the foveal to the parafoveal area. The optimal temporal frequency in Vh4 (an area on the ventral occipital cortex adjacent to area V3) [139] was identical to that between the foveal and parafoveal region. Neural activity in area Vh4 does not change in relation to eccentricity as is the case in the early visual stream [140].

In brief, motion-selective cells were found in areas V1, V2, V3, V3A, MT/V5, V3A, V2, VP [109,134]. Contrast selective cells in humans were present in areas V1 to V4 of the ventral stream, and in V3 and V3A and the intraparietal sulcus in the dorsal stream [137,138]. Cells in areas V1, V2, V3, V3A and Vh4 preferred high temporal frequency stimuli [139].

## 6. Is DD Due to Impairment of M-Cells?

Diminished visually evoked potentials to rapid, low-contrast stimuli have been reported in people with dyslexia, whereas responses to slow or high contrast stimuli were normal [49]. The results were interpreted as a defect in the LGN or area V1. A comparison of autopsy sections of the LGN of dyslexic and normal readers (mean age: 27 and 26 years respectively) showed smaller cell bodies in the magnocellular layers of the LGN of dyslexic readers. The cells in the parvocellular layers did not differ between the two groups. The magnocellular layers were, on average, approximately 27% smaller in the LGN of dyslexic readers and were also located in the parvo- and konio-cell layers in dyslexic readers. The difference between the size of the cell bodies was only on a statistical level of *p* < 0.05 [49]. Visually evoked potentials revealed that poor readers had significantly lower amplitudes and significantly shorter latencies elicited by the offset of low spatial frequency stimuli compared to normal readers [141]. The latencies of the early components of the visual evoked potentials of children with dyslexia were longer than those of normal readers only when the stimulus had a low spatial frequency on a steady background. There was no difference between poor and normal readers if the stimulus had a high spatial frequency. The authors concluded that the magnocellular visual pathway produces a slower response in children with dyslexia [142]. The latencies of the visual evoked potentials were longer in 35–56% of children with dyslexia compared to normal readers when moving high contrast stimuli were presented [143].

When subjects were asked to judge the velocity of coherently moving, low-contrast random dots that affect area MT/V5 of the magnocellular system, fMRI showed activation in normal readers, whereas this area was not activated in dyslexic subjects. When a stationary pattern stimulus was presented, normal and dyslexic readers exhibited activation in areas V1, V2, and the fusiform gyrus (Talairach coordinates (TC): x = between −58 and −38, y = between −32 and −48, z = between −6 and –18) [50]. The results revealed a visual processing deficit including the magnocellular system in dyslexic readers. In accordance with these results, there was only reduced activation in area V1 and area MT/V5, including adjacent motion-sensitive areas in dyslexic students compared with non-dyslexic controls in fMRI. A clear correlation was found between students’ ability to read words per minute and activity in areas V1 and MT/V5. There was higher activation of area MT/V5 than area V1, presumably because MT/V5 receives stronger M-fiber afferents than V1. These results underpin the assumption that poor reading is due to a dysfunction of the M-pathway including area MT/V5, which receives strong input from the M-fibers of the dorsal stream [51].

In a task in which children with dyslexia and normal readers were required to judge the direction of the movement of dots, children with dyslexia performed worse than children without dyslexia. Preschool children who were still pre-readers and showed poor performance in the dot movement task became poor readers at school age [38,144]. Flicker fusion thresholds were also lower in children with DD, including those with attention deficit hyperactivity disorder, than in non-dyslexic controls [145].

Abnormal M-cells were identified in areas V1 and MT/V5 of the dorsal stream and continued in the posterior parietal cortex, also belonging to the dorsal stream [146]. Therefore, it was assumed that M-cell pathology in the dorsal stream might play a key role in dyslexia [49,50,51,52,53,54,55].

The M-cells in the dorsal route that reach the parietal cortex were presumed to locate letters in a word, and direct attention and eye movements to letters. It was hypothesized that M-cells of the dorsal stream play an important role in the analysis of the order of letters in a word. However, M-cells have large visual fields and cannot identify the shapes or the features of small letters on the page of a book. The proponents of the magnocell theory of dyslexia assume that functionally impaired M-cells also disturb the visual perception of letters due to impaired saccadic suppression, defective binocular convergence and poor control of saccades. It is believed that this leads to the superposition of letters and the impression that letters are moving in the text [52,53,54].

The magnocell theory has also been criticized [147,148,149,150,151]. Based on their estimates of the frequency of magnocellular deficits, Skoyles et al. (2004) [147] concluded that the frequency of an impaired M-system is not higher in dyslexic readers than in normal readers. The frequency of magnocellular deficits in the latter may even be higher than that among dyslexic readers. Contemori et al. (2019) [150] reported that readers with DD had no higher thresholds in a grid detection task than controls for high temporal frequency (30 Hz) stimuli, as predicted by the magnocellular theory of dyslexia. This means that individuals with dyslexia were not impaired in a detection task when the stimuli elicited a typical magnocellular response. Their performance did not improve when the stimuli were mainly engaged in the parvocellular system. Therefore, the authors assumed that dyslexia is not exclusively due to a magnocellular deficit but that the parvocellular system may also be affected.

Hutzler et al. (2006) [151] found that children with dyslexia did not lack eye movement control, as advocated by the magnocell theory. When the children were asked to read a series of pronounceable pseudowords or search for two adjacent identical letters in a series of unpronounceable pseudowords, dyslexic readers performed more fixations than normal readers when reading pseudowords. There were no differences in fixation times or eye movements between dyslexic and normal readers, which confirmed that there was no deficit in eye movement control as assumed by the proponents of the magnocell theory of dyslexia.

When eye movements, fixation times, saccadic amplitude, and speech response latency were guided by a computer, children with dyslexia improved their reading performance drastically without any previous training [59]. In earlier studies, children with dyslexia reduced their reading mistakes by about two-thirds after learning a new reading strategy within less than 30 min [56,57,58,60]. This demonstrates that DD is caused by impaired eye movement control. Eye movement therapy did not take place at all or the reading therapy was too brief to successfully treat a disturbance of eye movement control.

In conclusion, the following can be assumed: it appears that a functional impairment of magnocells and parvocells of the visual system causes DD. There is, however, no sufficient evidence for the assumption that DD is only or predominantly caused by a magnocell deficit. Studies that show that dyslexic children exert appropriate eye movements when their eye movements are guided by a computer, and that they can learn to exert appropriate eye movements within 30 min, demonstrate that children with DD don’t suffer from a lack of eye movement control due to a magnocell deficit [56,57,58,59,60].

## 7. Is DD Due to an Impairment of Visual Processing?

The assumption that DD is caused by a dysfunction of the visual system, as posited by the magnocellular theory of DD, has been confirmed by studies on the influence of fixation times on the ability to simultaneously recognize pseudowords in children with dyslexia [57,58,59]. To prove a causal relationship between a stimulus parameter and reading performance, it must be demonstrated that the realization of this stimulus parameter improves reading performance in a reproducible way and that the absence of this parameter worsens reading performance. The studies by Werth (2018, 2019, 2021) [57,58,59] are the only studies in which the causal relationship has been examined in this way in a reproducible manner. To this end, pronounceable pseudowords with a length between 3 and 6 letters were displayed only once between 250 and 500 ms. Only 11 out of 200 children with DD aged between 8 and 15 years could recognize 6-letter pseudowords at a presentation time of 250 ms (Table 1). When the fixation time was prolonged to 500 ms, more letters in the pseudowords could be correctly recognized, and 17 children recognized all 6 letters simultaneously [57,58,59]. Of the 200 children with DD, 52 were unable to simultaneously recognize pseudowords that consisted of more than 3 letters even if the fixation times were prolonged up to 500 ms. Moreover, 132 out of 200 children with DD could not simultaneously recognize pseudowords that consisted of more than 4 letters if the fixation time was prolonged up to 500 ms [57,58,59]. When pseudowords were presented for such a short time that numerous reading errors occurred, the children could not see these words clearly enough. Under these tachistoscopic conditions, the stimulus was not presented long enough for the visual system to process it correctly. Reading mistakes occurred at all positions in the words. Letters were omitted, the positions of letters were swapped, letters were exchanged with other letters, and letters that did not occur in the word were inserted. When the children were asked whether they could see the words, they answered that the words appeared only for a very short time making it difficult for them to see all the letters clearly. All studies showed that the rate of reading errors was reduced to the extent that 95% of the words were read correctly when the fixation time was sufficiently increased [57,58,59] (Table 1). When the same pseudowords that were used in previous studies [57,58,59] were presented to 20 normal readers aged 19 to 20 years, and the experimental conditions were the same as in the previous experiments, they required a presentation time of only 50 ms to recognize 20 pseudowords consisting of 6 letters. This means that the number of letters that could be recognized simultaneously depended on the time the pseudowords were fixated, but there was also an individual maximum of letters that could be recognized simultaneously.

The improvement in the ability to recognize pseudowords simultaneously when the fixation time is prolonged is due to temporal summation [152,153,154,155,156,157,158,159,160,161,162,163,164,165]. Detection and recognition of visual stimuli [152,153,154,155,156,157,158,159], and visual acuity improve when fixation time increases [160,161,162,163,164,165]. This improved visual performance results from an increase in visual processing capacity as the fixation time is extended. When reading, the lateral geniculate, the striate, and extrastriate cortices of both cerebral hemispheres are activated [166,167,168,169,170,171]. The duration of stimuli results in increased activation of the primary visual and visual association areas and the left posterior and middle fusiform gyrus. The right middle fusiform gyrus and lingual gyrus do not react on the duration of the stimulus [168]. The temporal summation of visual stimuli is particularly pronounced in visual areas V1, V2, and V3 and is less pronounced in areas V4, VO (an area anterior to area V4 in the posterior inferior temporal sulcus) [172], in an anterior extrastriate area termed LO, area TO (an area in the occipitotemporal cortex that corresponds to area MT and MTS of macaques), and the intraparietal sulcus [173]. It was also detectable in the ventral temporal sulcus, which is part of the parvocellular pathway. The ventral temporal cortex is activated by sustained and transient, briefly displayed images. Neurons in the dorsal stream process rapid changes in the visual input. Neurons in the ventral temporal cortex process visual input that lasts for seconds as well as stimuli that are present for some ten ms [174]. We assume that the increased required fixation time and the resulting ability to recognize several letters at a time in children with DD is due to a dysfunction in visual areas V1, V2, and V3, and to a lesser extent to activation of areas V4 and IT, VO, LO, and TO via the parvocellular pathway. The prolonged fixation time required can be interpreted as a deficit of magnocells, which are particularly sensitive to rapid temporal changes. As parvocells account for at least 80% of foveal ganglion cells and have smaller receptive fields appropriate for analyzing higher spatial frequencies, we assume that parvo-cells also play a role in the required prolongation of fixation times. Both the magnocellular and the parvocellular pathways are most likely affected in readers with dyslexia.

The results of fMRI studies indicate that the limited ability to recognize only a small number of letters simultaneously is a deficiency in visual stimulus processing. When word length was varied between 3 and 9 letters, long words activated posterior visual areas more than short words [175]. fMRI also identified increased activity in the medial lingual gyrus of both hemispheres, the fusiform gyri, the right superior lingual gyrus and the medial cuneus. The response in the right medial lingual gyrus was stronger with low-contrast stimuli than with high-contrast stimuli. High-contrast stimuli and word length enhanced the response of the fusiform gyri in both hemispheres more than low-contrast stimuli. When the word length increased, activation in the lingual gyrus increased, but decreased when the contrast increased [175]. Schurz et al. (2010) [176] found a length effect for real words only in the occipital cortex, with strong activation in the lingual gyri of both hemispheres. A stronger word length effect was found in children with dyslexia than in age matched non-dyslexic children when 3- to 6- letter words and pseudowords were presented [177]. The authors concluded that children with dyslexia stick more to a sublexical reading procedure.

Taken together, experimental results [57,58,59] have shown that the ability to recognize several letters in a word or pseudoword simultaneously depends on the fixation time. Temporal summation [152,153,154,155,156,157,158,159,160,161,162,163,164,165] can explain the ability to recognize more letters in pseudowords simultaneously when the fixation time is prolonged. Regardless of the fixation time, there is an individual limit to the number of letters a person can recognize simultaneously. The question arises whether dyslexia is only a consequence of impaired visual cortex processing or whether brain structures that receive input from the visual cortex are crucial. This brought a region in the middle fusiform gyrus to the focus of the research.

## 8. The Role of the Fusiform Gyrus and the Visual Word Form Area

### 8.1. Anatomical Background

Warrington and Shallice (1980) [178] reported two patients who could only read letter by letter, but whose ability to recognize words as a whole was significantly reduced. The authors assumed that this type of acquired dyslexia is caused by damage to a neural network that enables the recognition of visual word forms. Cohen et al. (2000, 2002) [179,180] named an area in the ventral occipitotemporal cortex extending about 2 cm from the rostral to the caudal fusiform gyrus (Talairach coordinates (TC): x = −42,y = −57, z = −15) “visual word form area“ (VWFA). They assumed that this area plays a central role in prelexical processing and the identification of word forms. Cohen et al. (2003) [181] observed activation of the left and right VWFA upon presentation of a string of letters in a fMRI study. Activation of the VWFA was even stronger when a string of consonants was displayed. The left VWFA was more activated by a string of letters than by checkerboards in an fMRI study. The authors concluded that the VWFA is specialized for the processing of alphabetic stimuli.

The fusiform gyrus (FG) on which the VWFA is situated is a brain structure of the human ventral temporal cortex (Figure 2) that exists only in humans and is absent in the macaque brain [182]. The fusiform gyrus was subdivided into a posterior, a middle, and an anterior part. The exact location is indicated by the Talairach coordinates x = −39, y = −72, z = −18 for the posterior, x = −39, y = −60, z = −18 for the middle, and x = −39, y = −48, z = −18 for the anterior part of the fusiform gyrus [183]. This area includes the VWFA. Four cytoarchitectonic areas FG1, FG2, FG3, and FG4, were identified in the fusiform cortex [129,184]. FG2 and FG4 are cytoarchitectonically different. Layer III of FG2 has larger pyramidal cells than FG4 and a denser layer IV than FG4, whereas layer IV in FG4 is rather thin. Region FG3 is selective for places. Regions selective for faces and characters are located in FG2 (containing the visual word form area) and FG4. The results show that functional regions that are believed to be selective for characters are located in different cytoarchitectonic regions in the fusiform gyrus [185]. The inferior longitudinal fasciculus (ILF) connects the VWFA to the occipital cortex, and the arcuate fasciculus links the VWFA to the supramarginal gyrus which is regarded as a perisylvian language area [186].

### 8.2. Is There a Specialization for the Recognition of Words?

Cohen et al. (2000, 2002) [179,180] assumed that the VWFA is specialized for the recognition of visual words because some researchers [187,188,189,190] found higher activation of the VWFA in fMRI on real words or pronounceable pseudowords than on a succession of consonants or on a string of signs that were not letters (false fonts).

Polk et al. (2002) [187] reported in an fMRI study that in some subjects an area in the fusiform gyrus in the left hemisphere was more activated by strings of digits and strings of letters than by a fixation point and was more activated by strings of letters than by strings of digits. The presentation of words and pseudowords activated the left posterior (TC: −28, −88, −1 2; −28, −82, 0), the left midfusiform gyrus (TC: −28, −82, 0; −42, −54, −12), the right fusiform gyrus and the posterior lingual gyrus in both hemispheres. Letters exhibited greater activity than false fonts in the left lateral extrastriate cortex, the inferior occipital gyrus, the left anterior parietal lobe, and the left anterior and posterior fusiform gyrus [169].

An area in the posterior occipital temporal sulcus (OTS), corresponding to the VWFA [179,180,191], a region in the middle occipital temporal sulcus, which is more anterior to the posterior occipital temporal sulcus in the left hemisphere and a region in the inferior occipital sulcus were specific for characters. Peak selectivity for these regions was at about 2 Hz stimulus presentation rates with a decline at 4 Hz. The regions that preferred characters had the slowest temporal processing capacity [192]. In the study by Stevens et al. (2017) [190], the VWFA was more activated by words than by pseudowords. No other areas of interest discriminated similarly between words and pseudowords. The left and right fusiform face area discriminated more between pictures than the VWFA, which performed poorly in a picture discrimination task. The authors assumed that the VWFA is connected to the Wernicke area and is therefore specialized in processing real words.

Binder et al. (2006) [193] demonstrated that increasing familiarity with a sequence of letters resulted in increasing activation in the left lateral fusiform gyrus (TC: −44, −60, −12 and −41, −53, −7) with the main activation in the VWFA. Other authors reported poorer activation on highly familiar words [194].

Investigation of cerebral activation during reading and spelling using fMRI showed that reading and spelling enhanced activity in the left ventral occipitotemporal region, including the VWFA. Activity in the anterior supramarginal and postcentral gyri (dorsal stream) was correlated with reading but not with spelling [195]. In congenitally blind subjects, fMRI showed activation of the VWFA only when the subjects read real words in Braille but not when they were reading nonsense words in Braille. Therefore, the VWFA was considered a language-specific area regardless of the sensory modality of the word information [196].

In a meta-analysis Taylor et al. (2013) [197] identified stronger activation for real words than for pseudowords in the fusiform gyrus, but also in the left and right middle temporal gyrus, in the parahippocampal gyrus, the angular gyrus, and left medial orbitofrontal cortex. Consistent with the results of Taylor et al. (2013) [197] fMRI studies of adults without dyslexia conducted by Fischer-Baum et al. (2018) [198] found that words resulted in greater activation than non-words in the left and the middle occipital temporal cortex, in area V1, the right middle temporal gyrus, precuneus of both hemispheres, and the right supramarginal gyrus. They found three minor activation sites in the calcarine fissure, lingual gyrus, and superior occipital gyrus of the right cerebral hemisphere. Non-words elicited a strong response in the inferior frontal cortex, anterior insula, the medial region of the superior frontal gyrus of both hemispheres, and in the left inferior parietal lobe, including the angular gyrus, which was regarded as part of the lexical processing pathway. However, not all subjects tested by Fischer-Baum et al. (2018) [198] displayed equal activation in the left ventral occipitotemporal cortex (OT), the angular gyrus, and the inferior frontal gyrus.

Woolnough et al. (2021) [199] performed intracranial recordings while a string of non-letter signs, infrequent and frequent letter bigrams, quadrigrams, and real words were visually presented. A word-length effect was first registered after 75 ms at the occipital pole, which propagated anteriorly. The anterior areas of the ventral occipital temporal cortex responded less to a succession of non-letter signs and to a succession of infrequent letters, but were more responsive to real words, and showed a preference for high frequency words. The responses in the mid-fusiform cortex distinguished between an unpronounceable sequence of infrequent letters and real words. There was no difference between the responses to real words, bigrams, quadrigrams, or strings of frequent letters. In contrast, the occipitotemporal cortex responded more strongly to a succession of non letter signs than to words. The mid-fusiform cortex appeared to be specific for the identification of real words. It was the first region that responded to sublexical and lexical features, before the VWFA yielded a response. The magnitude and duration of the response in the mid-fusiform cortex depended on the frequency of words in natural language. The mid-fusiform cortex was regarded as the hub of the orthographic lexicon, and the long-term memory representations of visual word forms. The word-selective response propagated posteriorly from the mid-fusiform cortex to the posterolateral ventral occipito-temporal cortex within 500 ms.

### 8.3. No Priority for Words over Pseudowords

Other researchers found that the VWFA is not only activated by real words, but is equally activated by pseudowords and real words. In their meta-analysis, Jobart et al. (2003) [200] assumed that words, as well as non-words, may be processed within the VWFA in the left occipitotemporal cortex. In their review, the authors reported that there is evidence that the superior temporal areas, supramarginal gyrus, and the operculum of the inferior frontal gyrus are the brain regions that combine graphemes with phonemes. Corresponding to the demonstration of equal responses in the VWFA to words and pseudowords found in earlier studies, the VWFA activation to pseudowords and real words was similar when the pseudowords conformed with the orthographic rules of the subjects’ language [201,202].

Schurz et al. (2010) [176] also found that the VWFA on the left fusiform cortex responds equally to real words and short pseudowords. The anterior (MNI: −36, −42, −20; TC: −34, −43, −13) and middle (MNI: −44, −56, −22; TC: −42, −57, −14) fusiform cortex and the VWFA were activated by both words and pseudowords. Higher activation was observed for long words than for short words. The left occipital cortex (MNI: −26, −82, −18; TC: −26, −81, −11) was more activated by longer than by shorter pseudowords and by longer than by shorter words.

EEG recordings with intracerebral electrodes in the ventral occipital cortex of patients displayed letter-selective responses in an area of the left ventral occipital cortex. Neurons in the middle fusiforn gyrus, a region corresponding to the VWFA, were activated by words and pseudowords [203]. Baker et al. (2007) [189] reported that a region of the occipitotemporal sulcus and fusiform gyrus corresponding to the VWFA responded more to English words than to line drawings. There was also higher activation to English words than to Hebrew words that the subjects could not read, to strings of digits, and to Chinese characters, but not to strings of consonants. The activation to English words and consonant strings was equal. This shows that pronounceability of English words, in contrast to other stimuli, was not the decisive feature.

In some studies, the VWFA was even more activated by pseudowords than by real words [169,176,197,204,205,206,207,208]. Higher activity for pseudowords than for real words was documented in the occipitotemporal area, including the left fusiform gyrus (MNI: −48, −62, −12; TC: −46, −62, −5) and the left posterior fusiform gyrus (MNI: −36, −60, −12) (TC: −34, −60, −5) including the VWFA [197]. In an fMRI study, Binder et al. (2005) [206] found that the VWFA was even more activated by pseudowords than by regular and irregular words, whereas the angular gyrus was more activated by words than by pseudowords. These results were supported by Kronbichler et al. (2007) [207], who found that the VWFA (TC: −45, −48, −15; −45, −48, −15) was even more activated by pseudowords or pseudohomophones (i.e., words that sound the same but are spelled in a different and unusual way) than when real words were presented. Bruno et al. (2008) [209] corroborated these results. They also found that the occipital temporal area (TC: −44, −53, −13), which corresponds to the TC of the classical VWFA, was more activated by pseudohomophones and pronounceable pseudowords than by real words. The superior temporal gyrus and the inferior frontal gyrus were more activated by pseudowords than by pseudohomophones and were more activated by pseudohomophones than by real words. In agreement with Kronbichler et al. (2007) [207], the authors concluded that the occipital temporal region is a pivotal structure for reading printed words [209]. The reaction of the VWFA was not the same for all subjects. Some subjects showed equal activation for words and pseudowords in the VWFA (MNI: −46, −60, −12; TC: −44, −60, −5) whereas others showed greater activation for pseudowords than for real words [198]. This may indicate that orthographically familiar words affect only the posterior occipitotemporal regions, whereas orthographically unfamiliar pseudowords also affect the VWFA along the ventral visual pathway [210].

### 8.4. The VWFA Processes All Kinds of Stimuli

Several authors have questioned whether the VWFA is specific for the analysis of words. Words, legal letter combinations, orthographically illegal letter combinations, consonant strings, line drawings, and character strings in unknown writing exhibited BOLD activity in the VWFA. In their review, Price and Devlin (2003) [211] assumed that the VWFA is activated by visual word and non-word stimuli depending on task demands. The VWFA appears to be part of a system that processes visual stimuli such as words, pseudowords, false fonts, line drawings, and objects.

A region in the posterior VWFA showed BOLD activity for all stimuli [212,213]. The results contradicted the assumption that there is specialization for words or letters in the left occipitotemporal cortex or the VWFA. The VWFA appears to be involved not only in reading but also in processing similar types of stimuli. The VWFA and adjacent cortex are probably part of a neural network processing all kinds of complex visual stimuli [212,213]. The primary and secondary visual areas (Brodmann’s areas 17, 18 and 19) were not the only areas highly activated by all kinds of strings. The left middle fusiform gyrus, including the VWFA, was also equally activated by all kinds of strings such as pseudowords, unusual strings of letters, and false fonts in regions with the coordinates TC: −46, −56, −22; TC: −46, -52, −22; TC: 46, 54, 22; TC: −44, −56, −22. Unpronounceable strings of letters and false font strings resulted in the strongest activation in the inferior occipital cortex. It appeared that there is a progression from a more elementary processing of single letters in posterior areas to the processing of letter strings in more anterior areas [214].

Wright et al. (2008) [215] found that some subjects showed preference for pictures over words, and some showed a preference for written words over pictures so that there was no difference at group level in the occipital temporal region within a 5 mm radius around the TC: −42, −70, −14. The VWFA is a visual area is supported by the results of Kay and Yeatman (2017) [216], who found that activity in the VWFA increases with increasing contrast, which is a basic feature of visual stimuli that should be processed at an early stage.

It has been shown that the VWFA in the middle fusiform area reacts not only to words in familiar writing but also to a sequence of unknown foreign writing [217,218,219,220]. fMRI scans showed equally strong activation in the middle (TC: −39, −60, −18) fusiform gyrus and the VWFA, the posterior (TC: −39, −72, −18) fusiform gyrus, and the left inferotemporal gyrus (TC: −51, −48, −18) for English words and for a set of Korean characters that were unknown to the subjects [218]. A study in which strings of Chinese characters were displayed also demonstrated that the VWFA is activated by Chinese characters that were unknown to the subjects [217].

The medial surface of the left anterior fusiform gyrus was activated by real words and letter strings in unfamiliar foreign writing, and by pictures of objects [212,213,221,222,223,224,225]. Van Doren et al. (2010) [226] found responses to words and pictures in an area with TC: −51, −57, −15 in the fusiform gyrus. Kherif et al. (2010) [225] also found higher activation for picture naming than for reading with the highest activity in the left fusiform gyrus (MNI: −46, −62, −16 and −42, −36, −18) (TC: −44, −63, −9 and −39, −37, −11). Kay and Yeatman (2017) [216] showed that the VWFA is not specific for words, but reacts to stimuli such as faces. The face area in the fusiform gyrus also responds to visually presented words. The magnitude of a BOLD response in the VWFA depended not only on such stimulus features and on attention, but also on how difficult the task was for the subject. If these influences were small, the VWFA was also sensitive to low-level stimulus features. If the task became more difficult, activity in the VWFA also increased, presumably due to top-down influence. The authors assume that this top-down modulation is mediated by the inferior parietal sulcus via the vertical occipital fasciculus as described above. When pictures were presented once or repeatedly to adult subjects between 26 ms and 221 ms, fMRI showed not only activation of areas V1 and V2, ventral regions of area V4, and area LO, but also activation of the fusiform gyrus of both cerebral hemispheres [110]. The activated region was anterior to the fusiform face area, (situated somewhat anterior to the classical VWFA, TC on the the left fusiform gyrus: −34, −45, −16). Activation increased when recognition improved and when the subjects indicated that the pictures were seen more clearly. The authors assumed that the fusiform gyrus is not involved in the analysis of shapes that is completed in areas V1, V2, V3, V3a, and ventral area V4. Rather, it appears that the fusiform cortex is involved in establishing a connection between the visual image of an object and features of this object stored in memory [110]. In agreement with these results, an area on the fusiform gyrus (TC: −42, −63, −9) adjacent to the classical word form area displayed a strong response as soon as the adult subjects indicated that they recognized the pictures in a picture detection task [223].

The VWFA is connected to the language network including the STS and to frontal and parietal systems enclosing the IPS, which are assumed to be involved in attention. The connection between the VWFA and the fronto-parietal attentional system was stronger than the connection between the VWFA and the brain’s language system. Therefore, Chen et al. (2019) [227] assumed that the VWFA plays an important role in processing different visual stimuli other than words and that the VWFA plays an important part in monitoring attention in conjunction with the fronto-parietal attention system.

### 8.5. A Hierarchy of Visual Processing

It was proposed that letter strings are hierarchically processed in progression from areas V1 and V2 to V4 and then to the left fusiform gyrus. Areas V1, V2, V3 and V4 were assumed to process word length, visual contrast, rate, and duration, whereas letters, bigrams and morphemes are processed hierarchically in more anterior areas of the fusiform gyrus. The mid- and posterior regions of the fusiform gyrus were regarded as specific for the orthography of visual words, whereas the most anterior region of the fusiform was thought to be responsible for processing the semantic aspects of visual words [228]. Vinckier et al. (2007) [229] observed a gradient in selectivity in the occipitotemporal cortex and the inferior frontoinsular cortex for high-level written words. Activation in the occipitotemporal cortex became increasingly selective toward the anterior fusiform cortex. These gradients were more pronounced in the left hemisphere, indicating that the left occipitotemporal VWFA is not a homogeneous neural network, but has a hierarchical structure. When the visual stimuli were real words, the fusiform gyrus in the occipitotemporal cortex, anterior insula, intraparietal cortex of both hemispheres, the left prerolandic cortex, and the left supplementary motor area showed activation in the fMRI. The anterior ventral occipitotemporal cortex in the left hemisphere was more activated by real words and quadrigrams, whereas this region in the right hemisphere was more activated by false fonts. The left occipitotemporal region, including the VWFA, preferentially responded to words or orthographically correct pseudowords more than to infrequent and frequent letters and frequent bigrams [170,181,223]. When adult readers were exposed to false font strings, strings of infrequent letters, bigrams, quadrigrams and real words, increasing selectivity was found from posterior to anterior regions of the occipitotemporal cortex, with the highest activation for high-level stimuli in the anterior fusiform gyrus. These results support the assumption that the left occipitotemporal VWFA is an inhomogeneous structure that displays a hierarchically organized functional and spatial organization [230]. Whereas activation in the most posterior parts of the fusiform gyrus was equal for false fonts, pseudowords and real words, more anterior areas showed greater selectivity for stimuli more similar to real words. The selectivity increased in the anterior direction. The results of Vinckier et al. (2007) [229] were confirmed by Oulade et al. (2015) [230]. They also found a posterior to anterior gradient of increasing word selectivity in the left occipitotemporal cortex of children without dyslexia. No activity was found in these regions in children with dyslexia. A medial to lateral gradient of increasing word selectivity was found in the left inferior frontal cortex of nondyslexic children without dyslexia, but not in children with dyslexia. In contrast to children with dyslexia, children without dyslexia displayed a functional connection between the occipitotemporal cortex, which displayed selectivity for words, and the inferior frontal cortex [230].

Lerma-Usabiaga et al. (2018) [231] distinguished between a posterior and a middle region within the occipital temporal sulcus (OTS), which have different functions. The posterior OTS (corresponding to the posterior VWFA) may process low-level visual information [212,220,223]. The middle OTS (corresponding to the classical VWFA) [232] analyzes word forms and orthographic features. The posterior OTS corresponds to area FG2 and the middle occipital temporal sulcus corresponds to area FG4 of Weiner et al. 2017 [184,185]. These studies support the assumption that the VWFA is neither specific for real words nor for a string of known letters, but may be involved in the processing of all kinds of shapes [211,212,213,225,228].

The question of whether the VWFA plays a role in DD has been addressed by some investigators by examining the excitability of the VWFA using mainly fMRI. In several fMRI studies, subjects with dyslexia showed reduced activation in the classic VWFA [233,234,235], and the adjacent anterior regions of the left fusiform gyrus compared to normal readers [236,237,238,239,240,241,242,243,244]. Reduced activation of the word form area that is part of the left ventral occipital cortex is a common finding in subjects with dyslexia [179,180,188,193,194,222,229,232,245,246,247,248,249,250]. However, the finding that readers with DD show reduced activity in certain brain structures does not support the conclusion that the activity of these brain structures is a necessary condition for reading. If the function of a brain structure that is necessary for reading is impaired due to a developmental disorder, other brain structures that play no or only a minor role in reading may also be affected by the developmental disorder, and may show reduced activity in functional MRI. The best way to explore the role of a brain structure is by comparing a patient’s performance before and after acquired cerebral damage.

Taken together, the results of the studies on the function of the VWFA is that their neural networks are activated by real words, pseudowords, pseudohomophones, strings of foreign letters unknown to the reader (Amhari, Chinese, Korean, Hebrew), false fonts, line drawings, and pictures [211,212,213,214,215,216,217,218,219,220]. These results demonstrate that the VWFA receives visual input but it does not demonstrate whether the function of the VWFA is a necessary or sufficient condition for normal reading. This can only be demonstrated by the results of damage to the VWFA.

### 8.6. Reading after Damage to the VWFA

Beauvois et al. (1979) [251] reported the case of a patient who suffered from a lesion of the left angular gyrus, the posterior part of the middle temporal gyrus, and the inferior longitudinal fasciculus. He could understand and repeat oral language, visually identify letters in non-words, visually compare letters, and visually compare pseudowords, showing that he did not suffer from an impairment to visually process words. The patient could read real words by recognizing them “by means of global perception”, whereas his pseudoword reading was highly defective. It was assumed that the patient suffered from an impairment of the phonological reading process that adds phonemes to graphemes, while the visual perception of written language and understanding and expression of language were preserved. The authors concluded that the lexical reading process was sufficient, and the phonological reading process was superfluous when meaningful words were presented. However, the phonological reading process is indispensable when the words are meaningless [251].

Behrman et al. (1998) [252] showed that letter-by-letter reading in patients with pure alexia is not an isolated disturbance of word recognition, but may be the consequence of an impairment in identifying all kinds of complex pictures. Hillis et al. (2005) [253] described 22 patients with hypoperfusion or infarct of the VWFA, but whose area of Wernicke and the angular gyrus were unaffected. They had no impairment in written word comprehension and could perform lexical decision tasks. However, the patients were impaired in oral reading and oral and written naming of pictures. Cohen and Dehaene (2004) [248] documented a patient who had undergone surgical removal of the left occipitotemporal regions, including the VWFA. Alexia resulted from the deafferentation of the left fusiform cortex, but the patient retained letter-by-letter reading ability. Cohen et al. (2003) [181] stated that letter-by-letter reading is a consequence of a lesion in the left VWFA. They hypothesized that, after a lesion in this area, the symmetrical region in the right hemisphere mediates letter-by-letter reading.

These findings were corroborated by Gaillard et al. (2006) [254]. An area in the occipitotemporal cortex, which included the occipital temporal boundary, the inferior occipital temporal gyrus, fusiform gyrus, and the VWFA, was surgically removed in a patient who had shown no reading impairment before surgery. This area was specifically activated by words in fMRI prior to surgery. After surgery, the patient could still comprehend and repeat oral language and could complete a dictation test flawlessly, but displayed longer verbal reaction times to words and made more reading errors. Reading mistakes increased with word length when a single word was presented. The patient also retained a letter by letter reading pattern. The view that letter-by-letter reading is due to a dysfunction of the VWFA was also shared by numerous other researchers [186,228,248,254,255,256].

The patient described by Tsapkini and Rapp (2010) [228] had a lesion in the mid-fusiform gyrus with TC between y = −15 and y = −65, which is anterior to the lesion of Gaillard’s patient. The patient could read pseudowords without error and within normal response times. Real words were also read correctly. His response time was longer than that of normal controls when irregular and low frequency words were displayed. His performance in a lexical decision task was no different from that of normal controls confirming that his word understanding was perfect irrespective of word length. His ability to associate phonemes with graphemes with phonemes was also undisturbed, showing that his letter and word recognition were normal. When the patient was asked to spell real words in a dictation, he made four times as many errors as healthy controls, whereas almost all pseudowords were spelled correctly. When real words were misspelled, the words were misspelled so that the sequence of sounds was approximately the same as that of correctly spelled words. It seems that either the memory for the correct spelling was disturbed or that the correct spelling could not be retrieved. The results indicate that functions situated in the mid-fusiform gyrus with TC between y = −15 and y −65 play a role in orthographic processing.

In contrast to the view that the left ventral occipitotemporal cortex, including the VWFA, is necessary for rapidly processing a string of letters at the same time and that a lesion in the VWFA results in letter-by-letter reading, a patient reported by Seghier et al. (2012) [257] had a lesion in the left occipitotemporal region, but was still able to read aloud short familiar words with short vocal reaction times. The short vocal reaction times excluded letter-by-letter reading and demonstrated that the letters of the words were recognized simultaneously. fMRI showed that in this patient the visual input to the occipital cortex was transmitted to the central part of the left superior temporal sulcus and to the left motor and premotor regions without involving the left ventral occipitotemporal cortex. The investigators assumed that his reading performance was accomplished using a reading pathway that involved a region in the left STS in the absence of the ventral occipitotemporal cortex. They concluded that the left STS pathway could mediate accurate reading of rapidly presented words.

Most studies [248,249,253,254,255] suggest that damage to the middle fusiform gyrus makes simultaneous recognition of multiple letters that make up a word difficult or impossible. This does not allow us to conclude that letter-by-letter reading is due to a diminished ability to associate phonemes with graphemes. Before such an association is achieved, the letters in the word must be seen simultaneously. When a word is displayed for such a short time that eye movements cannot shift one letter after another into the center of gaze, the image of all letters in the word reaches the retina at the same time. A word-length effect can be registered after 75 ms in the primary visual cortex [200]. This information is forwarded anteriorly. Readers are familiar with frequently occurring sequences of letters so that the images of these letter sequences match well-known letter sequences stored in visual memory. It can be assumed that visual processing of highly familiar images is facilitated compared to that of unfamiliar stimuli because of recurrent connections to the structures that process visual stimuli [111]. Reading familiar words is also easier than reading unfamiliar words or pseudowords because, apart from the shape of the word, only a few letters in familiar words need to be seen to deduce the whole word. To recognize a word, it is not sufficient to recognize only its shape as the term “word form” may suggest. Words (e.g., “nod” and “mob”) may have approximately the same shape, but the letters must also be recognized simultaneously to distinguish the words. Recognizing a word by its form means seeing the letters that make up the word at a time and seeing its shape. As described above, the temporal summation of visual stimuli improves the processing of unfamiliar letter sequences that make up pseudowords. The finding that there is an individual limit on how many letters children with dyslexia can recognize simultaneously, regardless of the fixation time, shows that letter sequences exceeding a given length are not sufficiently processed by the visual system and, therefore, cannot be seen completely.

The VWFA may also be involved in processing the holistic perception of different kinds of visual forms, including words. When patients read natural words and pseudowords letter by letter after damage to the VWFA [248,254], this indicates that the cerebral damage prevents the sequences of letters that make up pseudowords from being seen as a whole, or the sequences of letters are seen as a whole but the corresponding sequences of sounds cannot be retrieved from memory. To associate the word form of a pseudoword with a sequence of sounds, all letters in the pseudoword must be seen simultaneously. Therefore, it can only be claimed that a reading disorder is due to a disorder of the grapheme-phoneme association if the words to be read have actually been seen. Our studies [57,58,59] demonstrated that children with DD did not see the sequences of letters that made up the pseudowords completely.

If only pseudowords are read letter by letter, while real words are read as a whole after damage to the VWFA [257], this suggests that the shape of real words and some letters in real words must have been seen, and that incomplete visual processing nevertheless made it possible to deduce the whole word. When reading pseudowords, seeing the shape of the word and some letters is not sufficient to deduce the pseudoword. Each letter and its position in a word must also be seen. This may explain why subjects read pseudowords letter by letter, while real words can be read as a whole.

In summary, the finding that patients have difficulty reading pseudowords after damage to the VWFA suggests that the VWFA may have a role in the simultaneous recognition of a string of letters [186,228,248,254,255,256]. Real words can be recognized better because they can be deduced when only a few letters and the shape of the word are seen at a time.

## 9. The Neurobiological Basis of Grapheme to Phoneme Association

After letters and words have been processed visually, the sequences of letters must be connected with sequences of sounds. The prevailing hypothesis that DD is caused by a phonological deficit means that the neurobiological basis of DD consists of a dysfunction of the structures that associate seen sequences of letters with sequences of sounds. It has been suggested that grapheme-phoneme association is mediated by brain structures such as the inferior parietal lobe including the angular gyrus, the supramaginal gyrus, and the inferior parietal sulcus, the posterior superior, middle and inferior temporal gyrus, the anterior insula, the left inferior frontal cortex, the dorsal perisylvian area, and the fusiform gyrus. It was hypothesized that the connections between the left inferior parietal lobe, which encloses the angular gyrus, which has been assumed to integrate orthography and phonology [218] and the fusiform gyrus may be impaired in dyslexic readers. The function of the fusiform gyrus may also be impaired in dyslexic readers, and the left fusiform gyrus may have a weaker modulatory effect on the left inferior parietal lobule (containing the angular gyrus, the inferior parietal sulcus, and the supramarginal gyrus) in children with reading difficulties compared to controls [258,259,260,261]. This may be interpreted as a deficit in integrateing graphemes and phonemes. Graves et al. (2010) [194] found evidence that the grapheme-phoneme association is achieved by a distributed system including the left supramarginal, posterior middle temporal, and fusiform gyrus. They assumed that the superior temporal cortex and the inferior parietal cortex including the angular gyrus have a role in the grapheme-phoneme association [198]. Blau et al. (2009) [260] found an underactivation in the superior temporal cortex, which they considered the brain structure that integrates letters and speech sounds in dyslexic readers.

In contrast, Church et al. (2011) [208] assumed that the left supermarginal gyrus, but not the angular gyrus, is important for phonological processing. The temporoparietal cortex, including the angular gyrus and the superior temporal gyrus, was equally activated by phonologically identical words that were orthographically different and by words that were the same both phonologically and orthographically. The VWFA and the inferior frontal gyrus were more strongly activated by words that were orthographically different, but phonologically identical, than by words that were orthographically and phonologically identical. The authors concluded that the temporoparietal cortex is involved in phonological processing, and that the VWFA was involved in orthographic processing [261].

Numerous studies have shown that the frontal cortex is also important for the association between graphemes and phonemes. Besides the left posterior superior temporal or left posterior inferior temporal cortex, the anterior insula and frontal operculum, seem to be involved in grapheme-phoneme association [262] and phonological retrieval from memory [263]. Reduced activity in the the left inferior frontal, premotor, supramarginal gyrus, the left inferotemporal and fusiform gyrus was observed in dyslexic readers compared to normal readers when reading and when performing visuo-phonological tasks. This led to the hypothesis that these neural networks are preferentially involved in visual-to-phonology processes during reading. The dorsal left fronto-parietal stream was active during phonological tasks, but also appeared to play a role in visuo-spatial perception and attention. No cluster of activity was identified in area MT/V5 [264]. In agreement with this view, activation indicating phonological processing of written words was found in the ventral occipitotemporal cortex, inferior parietal lobule, frontal areas, and the angular gyrus, which may be included in phonological and semantic processing [198]. Dickens et al. (2019) [265] hypothesized that perisylvian areas are involved in sound–motor integration and that these areas are involved in phonological decoding which is assumed to depend on the function of dorsal perisylvian areas.

In agreement with the findings of Brunswick et al. (1999) [238] and Paulescu et al. (2001) [239], Mechelli et al. (2005) [266] found that pseudowords increased activation in the left posterior fusiform gyrus more than regular words at a low statistical threshold (*p* < 0.05). Activation in the posterior fusiform gyrus was correlated with increased activation in the dorsal premotor area. The authors assumed that the association between graphemes and phonemes is the result from a coupling between the posterior fusiform gyrus and the dorsal premotor cortex [266].

## 10. Summary and Conclusions

In our studies that yielded the same results in 3 repetitions [57,58,59], children with DD could not recognize multiple letters at short fixation times but could recognize multiple letters almost flawlessly at sufficiently long fixation times. This means that temporal summation was crucial for the simultaneous recognition of letters that make up words. The finding that recognition performance improved with longer fixation time argues against the assumption that reduced simultaneous recognition is due to reduced attention. If this were the case, one would expect that performance would deteriorate with a longer fixation time because subjects with reduced attention cannot maintain attention for a longer time. These studies have also shown that children with DD can recognize only a limited number of letters simultaneously even with long fixation times. When the fixation time increased the primary visual and visual association areas and left posterior and middle fusiform gyrus were activated. The highest increase in activation was registered in visual areas V1, V2, and V3 whereas areas V4, VO (an area in the posterior inferior temporal sulcus), LO, TO, and the intraparietal sulcus were less activated [110,151,152,153,154,155,156,157,158,159,160,161,162,163,164,171,172,174,175,176]. This indicates that the number of letters that can be recognized simultaneously depends on the duration of input to these brain structures. This is supported by the finding that posterior visual areas [175] and the lingual gyrus [176] responded more to longer than to shorter words.

The fact that a given brain region in fMRI shows a higher activation to certain stimuli than other brain regions does not mean that it is specialized in processing these stimuli. Higher activation of a brain structure may be because the task requires greater effort and attention from this brain structure. The decreased activity revealed by fMRI in given brain structures of readers with dyslexia does not necessarily mean that decreased reading performance is caused by decreased activity in these brain structures. In their meta-analysis, Maisog et al. (2008) [267] reported that normal readers showed greater activation than dyslexic readers in 96 foci. The regions where normal readers are likely to show greater activation than dyslexic readers contain the fusiform gyrus, superior temporal gyrus, precuneus, and inferior parietal cortex, including the angular gyrus, thalamus, and left inferior frontal gyrus of the left cerebral hemisphere. Weaker activation than in normal readers was found in the right fusiform, postcentral, and superior temporal gyri of dyslexic readers. We cannot assume that reduced activation in each of these 96 foci, detected in the fMRI, causes dyslexia. Even if activation of a brain structure is reduced in fMRI, the processing that occurs in that structure may still be sufficient for normal reading performance. The activation of a brain structure shows that it receives direct input from the retina and the LGN or from cortical areas that process visual stimuli. Even if a brain structure is activated by visual stimuli such as words it may nevertheless be dispensable for reading. Conversely, the reduced activity of a brain structure does not necessarily mean that the performance of this person is impaired. Reduced activity in a brain structure may be compensated for by other brain structures. Damage may be compensated, and visual performance may be preserved even after severe developmental damage to the occipital lobe [268,269]. However, in all post-rolandic cerebral lesions the visual field may be affected, and the visual field defect may interfere with reading abilities [270,271,272,273,274] (Appendix B).

It appears that the impaired ability to simultaneously recognize letter sequences results from reduced processing of visual stimuli in the primary and secondary visual cortex. There are similarities between reduced simultaneous recognition of the letters of a word and dorsal simultanagnosia for objects [275,276,277,278,279,280,281,282,283,284,285,286]. However, both disturbances are not identical (Appendix C).

Studies with patients who, after damage to the VWFA, did not recognize words as a whole, but read them letter by letter, suggest that the VWFA plays a role in the holistic recognition of words. If the reduced processing capacity of the fusiform gyrus containing the VWFA plays a role in DD, then this is presumably a visual role in stimulus processing during simultaneous recognition of letter strings. This is in agreement with the responses of the VWFA to visual stimuli. The VWFA is not specialized for processing words [211,212,213,225,228]. It cannot be assumed that a brain structure such as the VWFA, which is specialized for reading, has been developed. Reading and writing were common in the Roman Empire during the imperial period, but this ability disappeared with the fall of the Western Roman Empire. It was not until the 19th century that the ability to read and write became common again in Europe. Such a short existence of a literate population is not sufficient enough to develop a brain structure specific for reading. The brain uses structures already available to process complex visual patterns including the fusiform gyrus for visual word recognition.

The evidence that DD is not a phonological disorder, but a visual processing disorder, emphasizes the necessity to focus on therapies that help improve impaired visual processing and compensate impaired visual functions.

## Figures and Tables

**Figure 1 brainsci-11-01313-f001:**
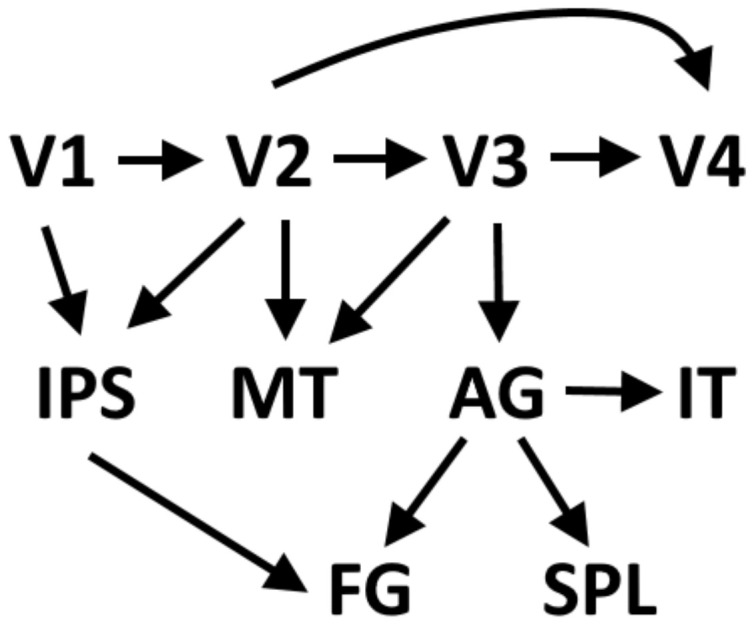
Forward connections of the visual cortex [125,126,127,128,129]. V1, V2, V3, V4: visual areas; IPS: inferior parietal sulcus; MT: medial temporal area; AG: angular gyrus; IT: inferior temporal area; FG: fusiform gyrus; SPL: superior parietal lobule.

**Figure 2 brainsci-11-01313-f002:**
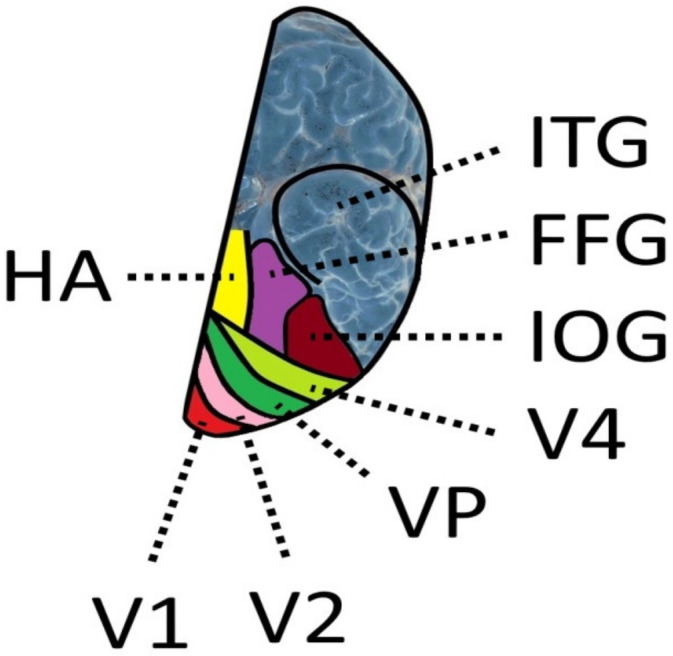
Ventral view of the left hemisphere [124,125,126,133]. V1, V2, Vp (corresponding to ventral area V3), V4: Visual areas. IOG: inferior occipital gyrus; FFG: fusiform gyrus; ITG: inferior temporal gyrus; HA: hippocampal area.

**Table 1 brainsci-11-01313-t001:** Pooled data of three studies [57,58,59] that show at which fixation times 200 children with dyslexia were able to read at least 95% of the pseudowords correctly. Top row: number of letters that made up the pseudowords. First column on the left: fixation times of pseudowords; second column: number of subjects who were able to read 3-letter pseudowords within fixation times between 250 and 500 ms; third column: number of subjects who were able to read 4-letter pseudowords within fixation times between 250 and 500 ms; fourth column: number of subjects who were able to read 5-letter pseudowords within fixation times between 250 and 500 ms. Fifth column: number of subjects who were able to read 6-letter pseudowords within fixation times between 250 and 500 ms.

Fixation Time Milliseconds	Number of Letters Recognized			
	3 Letters	4 Letters	5 Letters	6 Letters
	Number of Subjects Who Recognized ≥ 95% of the Pseudowords Correctly			
250 ms	24	30	28	11
300 ms	7	4	5	0
350 ms	4	9	11	0
400 ms	9	20	3	6
450 ms	5	1	1	0
500 ms	4	4	3	0
∑Subjects	52	80	51	17

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
