# Peer review of "Is Developmental Dyslexia Due to a Visual and Not a Phonological Impairment?"

_brainsci, 2021, doi:10.3390/brainsci11101313_

Round 1
Reviewer 1 Report
Teh Review represented an interested point of view of neurological basis of teh Developmental dyslexia
Teh author defines reading disorders, like all cognitive disorders, according to teh criteria of teh Diagnostic and Statistical Manual of Mental Disorders (DSM5) as dyslexia, indicated by (1) inaccurate and effortful word reading, (2) difficulty understanding teh meaning of what is read and (4) difficulty with spelling that persisted for at least six months and remains below teh skills expected for teh chronological age. Teh author attempted to confirm all claims on this definition in his review.
Teh author developed a computer strategy that controls how many letters subjects attempted to recognize simultaneously, teh fixation times, teh eye movements, and teh speech onset latency. This paradigm (pseudowords and words experiments) manipulates teh performance along two routes (lexical and sublexical).
However, surprisingly teh author used a title as “Developmental dyslexia is due to a visual and not to a phono logical impairment”.
Based on neuroscience research, teh neuronal mechanisms underlying teh different cognitive dyslexic types with and without phonological deficits (Heim et al., 2008) treat teh phonological processes as separate, specific sources of disability in developmental dyslexia (Van Ermingen-Marbach et all., 2013). A study (Heim et al., 2010) of phonological awareness and reorientation of attention found various left and right frontal effects on brain activation. Such atypical neural mechanisms confirm that teh phonological deficit in dyslexia is not necessarily a result of teh disturbed magnocellular function (Van Ermingen-Marbach et all., 2013). Increased activation in teh left inferior frontal gyrus of children with dyslexia has been observed to compensate for their dysfunction in teh left posterior regions (posterior part of superior temporal gyrus, teh supramarginal, and angular gyri of teh left inferior parietal lobe). Some authors (De Vos et al., 2017) claim that teh increase in teh neural synchronization to phoneme rates in dyslexia correlates with later shown phonological and reading skills that might be attributable to specific experience-driven changes in teh auditory cortex. Other authors show an increase in teh neural synchronization to phoneme rates in controls (Lehongre et al., 2011). Systematically atypical neural activity has been observed in children with developmental dyslexia (Saribay et all., 2017). Their speech encoding was explained by teh temporal sampling theory in developmental dyslexia (Goswami, 2011). Some authors (Power et al., 2016) claim that teh increase in teh neural synchronization to phoneme rates in dyslexia correlates with later shown phonological and reading skills that might be attributable to specific experience-driven changes in teh auditory cortex.
Based on these additional literature could you consider teh title or discuss teh phonologically based arguments and theories?
Van Ermingen-Marbach, M.; Grande, M.; Pape-Neumann, J.; Sass, K.; Heim, S. Distinct neural signatures of
cognitive subtypes of dyslexia with and without phonological deficits. NeuroImage Clin. 2013, 2, 477–490.
Heim, S.; Tschierse, J.; Amunts, K.;Wilms, M.; Vossel, S.;Willmes, K.; Grabowska, A.; Huber,W. Cognitive
subtypes of dyslexia. Acta Neurobiol. Exp. 2008, 68, 73–82.
De Vos, A.; Vanvooren, S.; Vanderauwera, J.; Ghesquière, P.;Wouters, J. A longitudinal study investigating
neural processing of speech envelope modulation rates in children with (a family risk for) dyslexia. Cortex 2017, 93, 206–219.
Lehongre, K.; Ramus, F.; Villiermet, N.; Schwartz, D.; Giraud, A.-L. Altered Low-Gamma Sampling in Auditory Cortex Accounts for teh Three Main Facets of Dyslexia. Neuron 2011, 72, 1080–1090.
Heim, S.; Grande, M. Fingerprints of developmental dyslexia. Trends Neurosci. Educ. 2012, 1, 10–14.
Heim, S.; Grande, M.; Pape-Neumann, J.; Van Ermingen, M.; Me_ert, E.; Grabowska, A.; Huber, W.; Amunts, K. Interaction of phonological awareness and ‘magnocellular’ processing during normal and dyslexic reading: Behavioural and fMRI investigations. Dyslexia 2010, 16, 258–282.
Saribay, S.; Noble, H.L.; Goswami, U. Neural Entrainment and Sensorimotor Synchronization to teh Beat in Children with Developmental Dyslexia: An EEG Study. Front. Neurosci. 2017, 11, 360.
Power, A.J.; Colling, L.J.; Mead, N.; Barnes, L.; Goswami, U.C. Neural encoding of teh speech envelope by
children with developmental dyslexia. Brain Lang. 2016, 160, 1–10.
Goswami, U. A temporal sampling framework for developmental dyslexia. Trends Cognit. Sci. 2011, 15, 3–10.
Teh text needs same minor clarifications / corrections:
- Teh sentence on line 189 needs to be clarified:
189: “Werth (2007) [112] demonstrated that humans can locate auditory stimuli and exert eye- and head movements to teh stimulus in teh absence of teh cortex and white matter of both cerebral hemispheres.”
- Teh sentence on line 267 needs to cite a literature:” V2, V3 and V4 there are stripes of color selective cells.”
- Teh sentences must be clearly explained: 281-283: Neural activity in area hV4 does not change in relation to eccentricity as is teh case in teh early visual stream [134]. Friderikson et al. (2016) [135] found that teh dorsal stream is more involved in phonological production tasks and that teh ventral stream predominates in lexical-semantic comprehension tasks.
- Are you talking about developmental dyslexia or dyslexia in adults?
288- 292: A comparison of autopsy sections of teh LGN of dyslexic and normal readers showed smaller cell bodies in teh magnocellular layers of dyslexic readers´ LGN. Teh cells of teh parvocellular layers did not differ between teh two groups. Teh magnocellular layers were on average about 27% smaller in teh dyslexic readers´ LGN and were also located in parvo- and konio-cell layers in dyslexic readers.
- Why do you have subjects on age 15 years? Its symptoms are different with age. They have higher reading experience and they could recognize multiple letters at short fixation times. If they still have a reading deficit, this dyslexia at this age is no longer developmental dyslexia. It is already dyslexia on young adulthood.
“374: Only 11 out of 200 children with DD aged between 8 and 15 years were able to recognize 6-letter pseudowords at a presentation time of 250 ms (Table 1). [25-27] “
Teh few questions arise from teh next teh cited literature [25-27], where teh author claims that
“Teh reading performance improved solely due to a reduction in segment size, an increase in fixation time, an extension of phonemic retrieval time, and an adaptation of teh amplitudes of teh saccades to teh size of teh segments. These were sufficient conditions to improve reading performance. This was regardless of whether teh segment was a syllable [25].
It was shown [26] that misread letters occurred at all locations in pseudowords, regardless of teh word´s length. Inadequate fixation, excessively large saccadic amplitudes, reduced ability to simultaneously recognize a sequence of letters, a longer required fixation time and a longer required speech onset latency were all identified as causes of dyslexia. Each of teh studies included in teh meta-analysis were much more efficient TEMPthan conventional therapeutic methods of reading disorder. Reading is a complex skill that requires many different brain functions. Teh gaze must be directed approximately at teh middle of teh word or word segment that must be read so that as many letters of teh word as possible are projected into teh area of teh retina, which has a sufficiently high visual acuity for reading. Teh field of attention has to be extended such that attention is directed to all letters that must be recognized. Teh word or word segment has to be fixated for a sufficiently long time such that teh pattern and arrangement of teh letters, their size, and their position within teh word can be processed by teh visual system. In addition, teh shape and position of teh letters of a word cannot be processed letter by letter. Teh pattern and position of a sequence of letters must be recognized almost simultaneously, which requires a separate brain capacity for simultaneous recognition (Poppelreuter, 1917;Werth, 2001, 2018). Teh simultaneous recognition of a sequence of letters is, however, a visual task that is made more difficult by lateral masking, also non as “crowding effect” (Stuart & Burian, 1962; Stromeyer & Julesz, 1972; Strasburger et al., 1991). Teh crowding effect reduces teh ability to recognize a letter as teh letter to be read is flanked on both sides by other letters. Visually processed letter sequences must be associated with learned sound sequences and meanings stored in teh memory. Finally, teh words that have been read must be stored in memory and combined into sentences. A deficit in one or more of these abilities, required for reading, may cause a reading deficit. It is unclear whether abnormal eye movements are also a cause of dyslexia or whether abnormal eye movements are a consequence of other reading impairments.
A computer-guided reading strategy [27] controls saccade amplitudes that exceed teh number of letters that teh reader is able to recognize simultaneously, and 4) an extended time needed to retrieve teh phonemes corresponding to teh graphemes from memory along with a premature onset of teh pronunciation of teh word, and a premature onset of a saccade to teh next word or word segment.
It is crucial for teh training that teh amplitudes of teh saccades that a reader has to perform are determined by teh number of letters that a reader can recognize simultaneously within a fixation phase.
Werth (2001) described teh role of a diminished ability to simultaneously recognize several letters in a word and its connection with teh saccade amplitudes, necessary for reading. Therefore, an increase of reversions does not necessarily prevent teh improvement of reading performance.”
- Werth, R. Rapid improvement of reading performance in children with dyslexia by altering teh reading strategy: A novel approach to diagnoses und therapy of reading deficiencies. Restor Neurol Neurosci 2018, 36, 679-691. doi: 10.3233/RNN-180829. 1080
- Werth, R. What causes Dyslexia? Identifying teh causes and effective compensatory therapy. Restor Neurol Neurosci 2019, 37, 591-1081 608. doi: 10.3233/RNN-190939. 1082
1083
- Werth, R. Dyslexic readers improve without training when using a computer-guided reading strategy. Brain Sci 2021, 11, 526. 1084 doi.org/10.3390/brainsci 11050526. 1085
610:
How were teh pseudowords created? What is teh frequency of use of teh words, used in teh experiments? Teh experiments consist of single words, which were into teh foveal area of teh retina (teh gaze must be directed at teh middle of teh word), what is teh case when there is a sentence? What happens with teh “crowding effect” in teh sentence?
- 4. Does teh VWFA process all kinds of (visual) stimuli?
- Could you discuss teh next citation?
733 -734: “readers with DD show reduced activity in certain brain structures does not support teh conclusion that teh activity of these brain structures is a necessary condition for reading.”
- Could you discuss teh results on teh 795-801?
“Teh investigators assumed that his reading performance was accomplished using a reading pathway that involved a region in teh left superior temporal sulcus (STS) in teh absence of teh ventral occipito-temporal cortex. They concluded that teh left STS pathway can mediate accurate reading of rapidly presented words.”
- There were no neurological mechanisms of teh eye movement deficits (807-829). In teh context of this paragraph would you briefly describe neurologically teh eye movement deficits with teh deficits in teh different parts of inferior occipital temporal cortex, especially of teh VWFA?
Is this last part of paragraph 8.6 more appropriate to be included to teh next paragraph 9?
- Clarify teh sentence on line 962: “Teh short existence of a literate population is not sufficient to develop a brain structure specific for reading.”
- Is teh next paragraph (965-971) contravened to teh title?
“There is also no evidence that teh fusiform gyrus is teh brain structure that connects graphemes to phonemes. It can only be claimed that DD is caused by a phonological disorder which consists in teh inability to associate letter strings with sequences of sounds if it has been proven that dyslexic readers see teh letter strings clearly but are unable retrieve teh correct sequences of sounds from memory. It is questionable whether teh connection between graphemes and phonemes is accomplished by only one single brain area [259] or whether it is accomplished by teh interaction of several distributed and interconnected brain structures [263].”
- What are teh arguments to include teh part 11.2 (Developmental dyslexia and hemianopic reading disturbance)? This part (line 974-975) should be pointed to consider patient cases:
“Teh correlation between reduced reading abilities and cerebral lesions must be interpreted with caution, as acquired cerebral damage is usually not restricted to anatomical or functional structures.”
Should be simultanagnosia here too? A dorsal simultanagnosia for objects is a rare but severe neurological disorder. Teh author has written for lesions. However, teh developmental dyslexia is based on immature functional and structural connectivity or brain networks. Does teh text really need this kind of comparisons? These lesions could result from a stroke or traumatic brain injury or old-age degenerative disorders.
After reading this review teh main question which left in mine mind is: “Is teh developmental dyslexia a clinical case or it is problem of teh child development? Can we help these children with strategies stimulating maturity or maturated processes?”
It is very difficult to distinguish these questions especially when there is not information for a hereditary burden and reading expirience.
- On line 1010, teh author claims for teh patients, that “All these reading mistakes disappeared when teh fixation time was sufficiently prolonged [25-27].” Teh literature (25-27) is not for patients.
Author Response
Referee 1
Teh Review represented an interested point of view of neurological basis of teh Developmental dyslexia
Teh author defines reading disorders, like all cognitive disorders, according to teh criteria of teh Diagnostic and Statistical Manual of Mental Disorders (DSM5) as dyslexia, indicated by (1) inaccurate and effortful word reading, (2) difficulty understanding teh meaning of what is read and (4) difficulty with spelling that persisted for at least six months and remains below teh skills expected for teh chronological age. Teh author attempted to confirm all claims on this definition in his review.
Teh author developed a computer strategy that controls how many letters subjects attempted to recognize simultaneously, teh fixation times, teh eye movements, and teh speech onset latency. This paradigm (pseudowords and words experiments) manipulates teh performance along two routes (lexical and sublexical).
However, surprisingly teh author used a title as “Developmental dyslexia is due to a visual and not to a phono logical impairment”.
Based on neuroscience research, teh neuronal mechanisms underlying teh different cognitive dyslexic types with and without phonological deficits (Heim et al., 2008) treat teh phonological processes as separate, specific sources of disability in developmental dyslexia (Van Ermingen-Marbach et all., 2013). A study (Heim et al., 2010) of phonological awareness and reorientation of attention found various left and right frontal effects on brain activation. Such atypical neural mechanisms confirm that teh phonological deficit in dyslexia is not necessarily a result of teh disturbed magnocellular function (Van Ermingen-Marbach et all., 2013). Increased activation in teh left inferior frontal gyrus of children with dyslexia has been observed to compensate for their dysfunction in teh left posterior regions (posterior part of superior temporal gyrus, teh supramarginal, and angular gyri of teh left inferior parietal lobe). Some authors (De Vos et al., 2017) claim that teh increase in teh neural synchronization to phoneme rates in dyslexia correlates with later shown phonological and reading skills that might be attributable to specific experience-driven changes in teh auditory cortex. Other authors show an increase in teh neural synchronization to phoneme rates in controls (Lehongre et al., 2011). Systematically atypical neural activity has been observed in children with developmental dyslexia (Saribay et all., 2017). Their speech encoding was explained by teh temporal sampling theory in developmental dyslexia (Goswami, 2011). Some authors (Power et al., 2016) claim that teh increase in teh neural synchronization to phoneme rates in dyslexia correlates with later shown phonological and reading skills that might be attributable to specific experience-driven changes in teh auditory cortex.
Based on these additional literature could you consider teh title or discuss teh phonologically based arguments and theories?
Van Ermingen-Marbach, M.; Grande, M.; Pape-Neumann, J.; Sass, K.; Heim, S. Distinct neural signatures of
cognitive subtypes of dyslexia with and without phonological deficits. NeuroImage Clin. 2013, 2, 477–490.
Heim, S.; Tschierse, J.; Amunts, K.;Wilms, M.; Vossel, S.;Willmes, K.; Grabowska, A.; Huber,W. Cognitive
subtypes of dyslexia. Acta Neurobiol. Exp. 2008, 68, 73–82.
De Vos, A.; Vanvooren, S.; Vanderauwera, J.; Ghesquière, P.;Wouters, J. A longitudinal study investigating
neural processing of speech envelope modulation rates in children with (a family risk for) dyslexia. Cortex 2017, 93, 206–219.
Lehongre, K.; Ramus, F.; Villiermet, N.; Schwartz, D.; Giraud, A.-L. Altered Low-Gamma Sampling in Auditory Cortex Accounts for teh Three Main Facets of Dyslexia. Neuron 2011, 72, 1080–1090.
Heim, S.; Grande, M. Fingerprints of developmental dyslexia. Trends Neurosci. Educ. 2012, 1, 10–14.
Heim, S.; Grande, M.; Pape-Neumann, J.; Van Ermingen, M.; Me_ert, E.; Grabowska, A.; Huber, W.; Amunts, K. Interaction of phonological awareness and ‘magnocellular’ processing during normal and dyslexic reading: Behavioural and fMRI investigations. Dyslexia 2010, 16, 258–282.
Saribay, S.; Noble, H.L.; Goswami, U. Neural Entrainment and Sensorimotor Synchronization to teh Beat in Children with Developmental Dyslexia: An EEG Study. Front. Neurosci. 2017, 11, 360.
Power, A.J.; Colling, L.J.; Mead, N.; Barnes, L.; Goswami, U.C. Neural encoding of teh speech envelope by
children with developmental dyslexia. Brain Lang. 2016, 160, 1–10.
Goswami, U. A temporal sampling framework for developmental dyslexia. Trends Cognit. Sci. 2011, 15, 3–10.
Author`s response: I added the following text and (lines 11-24) and parts of the introduction have been reworded: „A widely held opinion is that DD is a phonological disorder consisting of an impaired ability to associate letter sequences with the correct sound sequences [2-16]. Phonological awareness enables children to learn the grapheme-phoneme correspondence and to use it when reading or spelling. Phonological awareness does not designate one single ability, but comprises different abilities that are assumed to promote reading skills [11], such as decomposing words into syllables and sounds [12-18], identifying phonems in words [19-21], naming letters, objects, numbers and colors [15, 22], and rhyming [23]. An impairment of these abilities may coexist with DD, but it has not been demonstrated that there is a causal relationship between them and DD. Many correlations between DD and other impairments [29-52] have been reported without demonstrating a causal relationship. These correlations neither explain whether DD is the consequence of other performance impairments nor do they contribute to revealing the neurobiological causes of DD. Heim et al. (2008) distinguished between different kinds of DD based on different other impairments coexisting with DD. However, that DD and various other impairments exist simultaneously does not mean that these impairments cause different kinds of reading problems. The reading problems may be causally independent of other coexisting impairments as long as no causal relationship is demonstrated. To demonstrate the existence of a causal relationship between a performance deficit and DD, it must be shown that the performance deficit is caused by a lack of at least one necessary condition for reading or that no sufficient condition for reading is established. These logical relationships can only be investigated if they are based on a precise concept of causation [53-56] (Supplementalt text 1).“
To show that dyslexia is caused by the inability to associate a sequence of sounds with a sequence of letters seen, it must be ensured that the sequence of letters was seen clearly and completely and that the sequence of sounds could not be retrieved from memory. The literature that I reviewed in the paper shows that this was not the case. From my point of view, it makes no sense to present theories of phonological awareness in detail and subsequently show that there is no evidence that DD is due to a phonological disorder. I have included a text about phonological awareness as the reviewer requested to make clear that decreased phonological awareness is a class of different impairments that may correlate with DD.
I also extended the supplemental text about causation to make the concept of causation more lucid:
„In scientific contexts, a causal relationship i soften described by „is due to“,is brought about", "is the result of", "is conditioned by" or "is accounted for" instead of "is caused by". The fundamental importance of the concept of causation or its paraphrases is already shown by the fact that the concept of dyslexia is defined in the DSM5 with the help of the concept of cause, described as „accounted for“. Reading difficulties "... are not better accounted for by intellectual disabilities, uncorrected visual or auditory acuity, other mental or neurological disorders, psychological adversity, lack in the proficiency in the language of academic instruction, or inadequate educational instruction ..." (DSM 5 2013, p. 67). So far, several proposals have been made to define the concept of cause in a scientifically exact way, which, however, are mathematically formulated and difficult to apply in scientific practice (Spohn, Pearl, etc.). Here we will use a concept that is easy to handle (For a set-theoretical exact definition of the concept of cause see: Werth 2010, p.159). If a person can read without errors, this means that all necessary and sufficient conditions for correct reading are established. If only one necessary conditions is missing, a person cannot read. These necessary conditions include that the light stimulus and the background have a certain luminance, that the refractive media of the eye are translucent, that the retina is functional and not scarred, that the optic nerve, the optic tract and the geniculo-striate projection are not interrupted and that the visual cortex is present and functional. If all the necessary conditions for reading exist, reading is still not possible unless at least one sufficient condition is also established. One such sufficient condition is that the light stimulus is displayed on a monitor. This is not a necessary condition because the stimulus can also be displayed on a perimetric bowl or on a screen. This means that there are several possibilities to present the light stimulus, at least one of which must be established. A stimulus can only be seen if all necessary conditions and at least one sufficient condition must be established. The absence of a necessary condition or the absence of all sufficient conditions for seeing a stimulus are causes for not seeing a stimulus.
If DD correlates with impaired abilities in other domains this does not mean that these impaired abilities are causes for DD. Let A be an ability that is necessary for normal reading. Then, a deterministic causal relationship between an impairment of A and DD can be demonstrated if and only if (1) DD is present whenever A is impaired, (2) DD disapears whenever A is unimpaired, or (3) DD disappears if A is impaired and if the impaired ability A is compensated by another ability B.
If an impairment of the ability A cannot be compensated by another ability, then an unimpaired ability A is a necessary condition of normal reading performance.
If an impaired ability A can be compensated by another ability, then an unimpaired ability A and an unimpaired ability B which can compensate the impairment of ability A are both sufficient but not a necessary conditions for normal reading performance.
In terms of experimental procedures this means that a causal relationship between DD and an impaired ability A exists (1) if it can be demonstrated that the DD disappears as soon as the impaired ability A is eliminated by a therapy and (2) it is ensured that only the impaired ability A which may cause the DD is eliminated, and that no other impaired ability C coexists with DD and is also eliminated by the therapy. In ealier experiments (Werth 2001) this was achieved by changing the reading strategy by a computer program without any training whereupon reading performance improved immediately. This guaranteed that no other impaired ability such as impaired attention was improved at the same time.
Reference „Heim, S.; Grande, M.; Pape-Neumann, J.; Van Ermingen, M.; Me_ert, E.; Grabowska, A.; Huber, W.; Amunts, K. Interaction of phonological awareness and ‘magnocellular’ processing during normal and dyslexic reading: Behavioural and fMRI investigations. Dyslexia 2010, 16, 258–282.“ Has been added as refernce number 148.
Teh text needs same minor clarifications / corrections:Teh sentence on line 189 needs to be clarified:
189: “Werth (2007) [112] demonstrated that humans can locate auditory stimuli and exert eye- and head movements to teh stimulus in teh absence of teh cortex and white matter of both cerebral hemispheres.”
Author´s response: The following text was added: „The optic nerves and the brain stem were preserved. If an acoustic stimulus of 84 db was presented 40 times to his left or right ear the child always turned his head towards the stimulus. When no stimulus was presented in controltrials no head movement occurred.“
- Teh sentence on line 267 needs to cite a literature:” V2, V3 and V4 there are stripes of color selective cells.”
- Author´s response: References [131, 132] have been added.
Teh sentences must be clearly explained: 281-283: Neural activity in area hV4 does not change in relation to eccentricity as is teh case in teh early visual stream [134]. Friderikson et al. (2016) [135] found that teh dorsal stream is more involved in phonological production tasks and that teh ventral stream predominates in lexical-semantic comprehension tasks.
Author´s response: This passage has been reworded, paragraph 4, lines 27-34. Contrast sensitivity in areas V1, V2, V3 increased when temporal frequency of the stimuli in the periphery of the visual field was high. This was not the case in hV4 where the temporal frequency tuning was almost the same between the fovea and near-periphery [134].
This text has been added on page 9, lines 5-11: „Fridriksson et al. (2016) investigated brain areas of involved in speach production. The study is based on MRI data and behavioral testing of 136 post stroke survivors. The authors distinguished between a dorsal frontoparietal stream and a ventral temporal-frontal stream. The dorsal stream inluded the pars opercularis and premotor areas, and posterior regions in the supramarginal gyrus. The ventral stream includes the lateral temporal lobe, the inferior parietal lobule, the uncinate fasciculus and the iinferior frontal lobe. Ventral regions were involved in lexical–semantic analysis of speach, whereas dorsal regions were involved in the phonological–motor processing of speech production.“
- Are you talking about developmental dyslexia or dyslexia in adults?
288- 292: A comparison of autopsy sections of teh LGN of dyslexic and normal readers showed smaller cell bodies in teh magnocellular layers of dyslexic readers´ LGN. Teh cells of teh parvocellular layers did not differ between teh two groups. Teh magnocellular layers were on average about 27% smaller in teh dyslexic readers´ LGN and were also located in parvo- and konio-cell layers in dyslexic readers.
Author´s response: (mean age: 27 and 26 years respectively) has been added.
Why do you have subjects on age 15 years? Its symptoms are different with age. They have higher reading experience and they could recognize multiple letters at short fixation times. If they still have a reading deficit, this dyslexia at this age is no longer developmental dyslexia. It is already dyslexia on young adulthood.
“374: Only 11 out of 200 children with DD aged between 8 and 15 years were able to recognize 6-letter pseudowords at a presentation time of 250 ms (Table 1). [25-27] “
Author´s response: In the studies (Werth 2006, 2018, 2021) all subjects suffered from the same DD. In older subjects DD was already diagnosed when they were second graders. The children were referred to my neuropsychological unit because earlier therapy attempts were not successful. Subjects aged between 10 and 15 years did not differ from the reading problems in younger subjects.Older children were often better readers than younger children, and they made the same kind of mistakes. Reading experience does not necessarily change the symptoms of DD. Subjects with severe DD are usually no keen readers and their reading experience is qiuite limited. They can only recognize a limited number of letters at a time, they need longer fixation times, the reading saccades don´t match the numbers of letters recognized at a time and they need rather long response times. When subjects with DD grow older their reading problem is often less severe then in first-, second-, or third graders because some older subjects compensate their reading problem to a certain extent. The underlying reading problem remains the same. DD that exists in children is not always lost and often remains until adult age.
Teh few questions arise from teh next teh cited literature [25-27], where teh author claims that
“Teh reading performance improved solely due to a reduction in segment size, an increase in fixation time, an extension of phonemic retrieval time, and an adaptation of teh amplitudes of teh saccades to teh size of teh segments. These were sufficient conditions to improve reading performance. This was regardless of whether teh segment was a syllable [25].
It was shown [26] that misread letters occurred at all locations in pseudowords, regardless of teh word´s length. Inadequate fixation, excessively large saccadic amplitudes, reduced ability to simultaneously recognize a sequence of letters, a longer required fixation time and a longer required speech onset latency were all identified as causes of dyslexia. Each of teh studies included in teh meta-analysis were much more efficient TEMPthan conventional therapeutic methods of reading disorder. Reading is a complex skill that requires many different brain functions. Teh gaze must be directed approximately at teh middle of teh word or word segment that must be read so that as many letters of teh word as possible are projected into teh area of teh retina, which has a sufficiently high visual acuity for reading. Teh field of attention has to be extended such that attention is directed to all letters that must be recognized. Teh word or word segment has to be fixated for a sufficiently long time such that teh pattern and arrangement of teh letters, their size, and their position within teh word can be processed by teh visual system. In addition, teh shape and position of teh letters of a word cannot be processed letter by letter. Teh pattern and position of a sequence of letters must be recognized almost simultaneously, which requires a separate brain capacity for simultaneous recognition (Poppelreuter, 1917;Werth, 2001, 2018). Teh simultaneous recognition of a sequence of letters is, however, a visual task that is made more difficult by lateral masking, also non as “crowding effect” (Stuart & Burian, 1962; Stromeyer & Julesz, 1972; Strasburger et al., 1991). Teh crowding effect reduces teh ability to recognize a letter as teh letter to be read is flanked on both sides by other letters. Visually processed letter sequences must be associated with learned sound sequences and meanings stored in teh mmory. Finally, teh words that have been read must be stored in memory and combined into sentences. A deficit in one or more of these abilities, required for reading, may cause a reading deficit. It is unclear whether abnormal eye movements are also a cause of dyslexia or whether abnormal eye movements are a consequence of other reading impairments.
Author´s response: The crowding effect makes reading more difficult compared to visual tasks where no crowding effect is present. The role of the crowding effect has already been discussed in my previous papers (references 25-27) and is not repeated in the present paper. The results of the experiments have shown, however, that letters in pseudowords that were flanked on both sides by letters were not more often misread than letters at the end of pseudowords that were not flanked on both sides. This is not an accidental effect found in only one study but it could be evidenced in all studies. When fixation time was extended more letters were correctly recognized at a time and 95% of the words were correctly recognized. This shows that fixation time was crucial for pseudoword recognition.
A computer-guided reading strategy [27] controls saccade amplitudes that exceed teh number of letters that teh reader is able to recognize simultaneously, and 4) an extended time needed to retrieve teh phonemes corresponding to teh graphemes from memory along with a premature onset of teh pronunciation of teh word, and a premature onset of a saccade to teh next word or word segment.
It is crucial for teh training that teh amplitudes of teh saccades that a reader has to perform are determined by teh number of letters that a reader can recognize simultaneously within a fixation phase.
Author´s response: In order to read a text without mistakes a recognized word segment must connect gaplessly to the next word segment to be read. If the ampöitude of reading saccades exceeds the number of letters that can be recognized simultaneously the is a gap between the the sequences of letters that can be recognized simultaneously to the detriment that letters in the gap are not recognized. In this way too large sacccades cause reading mistakes. If searching eye movements are exerted after a sequence of letters had been recognized eye movements may miss the correct goal where eye movements should move in order to recognize the next word segment to be read. This is another way in which eye movements can cause reading mistakes. To that end reading therapy must show the reader which eye movement s/he must exert and how many letters s/he shoud try to read simultaneously. Only when the number of letters that can be recognized simultaneously match the amplitudes of saccades flawless reading is possible.
The verbal response times that sublects with DD need is due to the time that is needed to process visual stimuli and to retrieve the sequence of sounds corresponding to the sequence of letters from memory. If an eye movement to the next word segment to be read occurs before the analysis of visual stimuli and retrieval from memory are completed when subjects are reading a text, reading mistakes are frequent. This can be explained by the assumption that the visual analysis and retrieval of sounds from memory are disturbed by the requirement to visually process the letters of the next segment. When the computer steers the time when the next eye movement should be innitiated the rate of reading mistakes is considarably reduced,
Werth (2001) described teh role of a diminished ability to simultaneously recognize several letters in a word and its connection with teh saccade amplitudes, necessary for reading. Therefore, an increase of reversions does not necessarily prevent teh improvement of reading performance.”
- Werth, R. Rapid improvement of reading performance in children with dyslexia by altering teh reading strategy: A novel approach to diagnoses und therapy of reading deficiencies. Restor Neurol Neurosci 2018, 36, 679-691. doi: 10.3233/RNN-180829. 1080
- Werth, R. What causes Dyslexia? Identifying teh causes and effective compensatory therapy. Restor Neurol Neurosci 2019, 37, 591-1081 608. doi: 10.3233/RNN-190939. 1082
1083
- Werth, R. Dyslexic readers improve without training when using a computer-guided reading strategy. Brain Sci 2021, 11, 526. 1084 doi.org/10.3390/brainsci 11050526. 1085
610:
How were teh pseudowords created? What is teh frequency of use of teh words, used in teh experiments? Teh experiments consist of single words, which were into teh foveal area of teh retina (teh gaze must be directed at teh middle of teh word), what is teh case when there is a sentence? What happens with teh “crowding effect” in teh sentence?
Author´s response: Children with DD who participated in reading therapy and children who did not participate in therapy read the same sequences of pseudowords. Pseudowords contained only pronounceable letter sequences that also occur in German everyday language. Whenever possible, the sequences of letters also had approximately the same shape. Each pseudoword appeared only once in each test and was never repeated.
- Does teh VWFA process all kinds of (visual) stimuli?
Author´s response: The VWFA appears to be part of a system that processes all kinds of visual stimuli such as words, pseudowords, false fonts, line drawings, and objects, as described in section 8.
- Could you discuss teh next citation?
733 -734: “readers with DD show reduced activity in certain brain structures does not support teh conclusion that teh activity of these brain structures is a necessary condition for reading.”
Author´s response: A causal relationship between brain damage and an impaired performance can only be demonstrated by showing that a cerebral function is a necessary condition for a performance and that the absence of this brain function results in an impaired performance. For example, the function of the visual cortex is necessary for conscious visual perception because damage to the occipital lobe results in cerebral blindness. When the function of the occipital lobe is restored by therapy, conscious vision is restored. This means that damage to the occipital cortex is sufficient for cerebral blindness. However, it is not necessary for cerebral blindness because cerebral blindness can also occur after bilateral damage to the geniculate body or tot he geniculo-striate projection.
Could you discuss teh results on teh 795-801?
“Teh investigators assumed that his reading performance was accomplished using a reading pathway that involved a region in teh left superior temporal sulcus (STS) in teh absence of teh ventral occipito-temporal cortex. They concluded that teh left STS pathway can mediate accurate reading of rapidly presented words.”
Author´s response: This passage has been reworded: In contrast to the view that the left ventral occipito-temporal cortex including the VWFA is necessary for rapidly processing a string of letters at the same time and that a lesíon in the VWFA results in letter by letter reading, a patient reported by Seghier et al. (2012) [254] had a lesion in the left occipito-temporal region, but was still able to read aloud short familiar words with short vocal reaction times. The short vocal reaction times excluded letter by letter reading and demonstrated that the letters oft he words were recognized simultaneously. MRI showed that in this patient the visual input to the occipital cortex was transmitted to the central part of the left superior temporal sulcus (STS) and to the left motor and premotor regions without involving the left ventral occipito-temporal cortex. The investigators assumed that his reading performance was accomplished using a reading pathway that involved a region in the left superior temporal sulcus (STS) in the absence of the ventral occipito-temporal cortex. They concluded that the left STS pathway can mediate accurate reading of rapidly presented words.
- There were no neurological mechanisms of teh eye movement deficits (807-829). In teh context of this paragraph would you briefly describe neurologically teh eye movement deficits with teh deficits in teh different parts of inferior occipital temporal cortex, especially of teh VWFA?
- Author´s response: No eye movement disturbances have been reported after damage to the VWFA or to the inferior occipital temporal cortex including the fusiform gyrus.
Is this last part of paragraph 8.6 more appropriate to be included to teh next paragraph 9?
Author´s response: Paragraph 8. 6. „Reading after damage to the VWFA“ deals with reading problems after damage to the VWFA. There is no evidence that damage to the VWFA impaires phonem to graphem association. Instead, the VWFA may have a role in simultaneous visual processing of a string of letters that make up a word. In contrast paragraph 9. „The neurobiological basis of graphem to phonem association“ is about graphem to phonem association which is no visual processing. Therefore the topica of both paragraphs are completely different.
Clarify teh sentence on line 962: “Teh short existence of a literate population is not sufficient to develop a brain structure specific for reading.”
- Author´s response: The following sentence has been added: Reading and writing were common in the Roman Empire during the imperial period, but this ability disappeared with the fall of the Western Roman Empire. It was not until the 19th century that the ability to read and write became common again in Europe.
- Is teh next paragraph (965-971) contravened to teh title?
“There is also no evidence that teh fusiform gyrus is teh brain structure that connects graphemes to phonemes. It can only be claimed that DD is caused by a phonological disorder which consists in teh inability to associate letter strings with sequences of sounds if it has been proven that dyslexic readers see teh letter strings clearly but are unable retrieve teh correct sequences of sounds from memory. It is questionable whether teh connection between graphemes and phonemes is accomplished by only one single brain area [259] or whether it is accomplished by teh interaction of several distributed and interconnected brain structures [263].”
Author´s response: This paragraph has been reworded:This shows that the VWFA may be regarded as an area that contributes to the processing of visual stimuli. It cannot be assumed that a brain structure such as the VWFA, which is specialized for reading, has developed. Reading and writing were common in the Roman Empire during the imperial period, but this ability disappeared with the fall of the Western Roman Empire. It was not until the 19th century that the ability to read and write became common again in Europe. Such a short existence of a literate population is not sufficient to develop a brain structure specific for reading. The brain uses structures already available to process complex visual patterns including the fusiform gyrus for visual word recognition. There is also no evidence that the fusiform gyrus is the brain structure that connects graphemes to phonemes.
What are teh arguments to include teh part 11.2 (Developmental dyslexia and hemianopic reading disturbance)? This part (line 974-975) should be pointed to consider patient cases:
Author´s response: The cases were referenced: Patients with retrorolandic brain damage may also suffer from a homonynous visual field defect (patients M, F and A reported by Cohen et al. 2003 [175]and patients JT, BA, JH reported by Starrfelt et al. 2009 [267]) which can influence the reading disorder.
“Teh correlation between reduced reading abilities and cerebral lesions must be interpreted with caution, as acquired cerebral damage is usually not restricted to anatomical or functional structures.”
Should be simultanagnosia here too? A dorsal simultanagnosia for objects is a rare but severe neurological disorder. Teh author has written for lesions. However, teh developmental dyslexia is based on immature functional and structural connectivity or brain networks. Does teh text really need this kind of comparisons? These lesions could result from a stroke or traumatic brain injury or old-age degenerative disorders.
Author´s response: DD is indeed due to developmmental functional impairments of seural networks. The question is the functional impairment of which networks cause the symptoms of DD. To adress this question we must find out which neural networks mediate which ability related to reading. The only way to do this is to examine the impairments that result from a lesion of the network in question. In humans this is possible by neuropsychological investigation of impairments that result from damage to this very brain structure. Therefore, neuropsychological studies of behavioral impairments following circumscibed cerebral lesions is the key for our understanding of the role that given neural networks play for reading.
After reading this review teh main question which left in mine mind is: “Is teh developmental dyslexia a clinical case or it is problem of teh child development? Can we help these children with strategies stimulating maturity or maturated processes?”
It is very difficult to distinguish these questions especially when there is not information for a hereditary burden and reading expirience.
Author´s response: My experience from the examination of numerous children and adolescents with dyslexia in clinical practice is that without therapy, reading problems usually persist into adulthood. The results of my published studies show that DD seems to be caused by a dysfunction of the brain´s visual network. For some children, we can improve the ability to recognize multiple letters simultaneously with daily training, although this can take several months. Other children show no improvement. This depends, as we have also shown in the therapy of cerebrally blind areas, on how severely the brain structures are affected and how high the plasticity of the brain is. This plasticity can vary widely among children. The crucial problem, however, is that with school children you can't take your time with the therapy. Reading performance must be improved as quickly and effectively as possible. This was achieved with the therapy I developed, so that reading could be improved in a very short time. This is only possible by compensating for the impairment that hinders reading. At the same time, we try to improve the impairements (e. g. the impairmed ability to simultaneously recognize a string of letters) that are the cause of the reading disorder. However, this is time-consuming and comes up against individual limits.
- On line 1010, teh author claims for teh patients, that “All these reading mistakes disappeared when teh fixation time was sufficiently prolonged [25-27].” Teh literature (25-27) is not for patients.
- Author´s response: I am not sure if I have got the point you are making here. The papers with reference numbers 25-27 report the diagnostic and therapy with the computerprogram that we developed. The results of the pseudoword experiments that were completed in these studies are summarized in Table 1 in the present paper.
Reviewer 2 Report
In this special issue of MDPI’s Brain Sciences on “Neurobiological Basis of Developmental Dyslexia” the guest editor invites to write a short review with the aim of select and summarize the most substantial contributions to the field that will help to come to a solid understanding of the etiopathological mechanisms at the basis of the disorder. Overall the information collected represents valuable evidence about the topic, but due to the non-homogeneous and non-systematic presentation of the contents (in some cases too detailed and in other too general, often repetitive and redundant), the manuscript becomes difficult and slow to read.
My first concern is about the length of the present work that might not fit the format requested, however, a detailed “Supplementary Information” section might be useful and could help to keep the manuscript in a short format.
Then, I have some concerns about the writing style and the organization of the sections. In particular, key critical points are:
- Aim of the review and Methods Section
The aim of the work is not extensively discussed, I suggest explaining the purpose of the author and the hypothesis at the basis.
A Methods section is missing. It is necessary to clarify what kind of review it is: narrative, systematic, scoping, gap analysis…? Then the author should detail how the selection of the paper has been done and which criteria of eligibility he followed.
- Structure
The author should provide a brief summary at the end of each paragraph, which allows the reader to easily understand the results concerning the topic treated in that paragraph. The results are reported as a list, an overview is lacking. A table would be useful in order to organize all the study results cited in the review, especially the controversial ones. In each paragraph, the author should organize the scientific literature in a more systematic way, e.g.: dividing studies for population (clinical, general population); age (children, adult); methodology used (EEG, ERP, fMRI…).
- Writing style
Many paragraphs (e.g.: 8.2, 8.3 and 8.4) should be summarized in order to assert clear conclusions (e.g.: Which kind of stimuli does the VWFA process?).
I also suggest citing more recent and relevant literature, it seems that the author comes to conclusions based mainly on his own studies. Providing better citations will help place the current work in better context of the scientific progress in this area over the last years.
In the text there are several typographical error or misprint (e.g.: lines 57,81,182,237, 451) and inconsistent formatting (e.g.: lines 100-102; 422-431; 731-732; 761-762), it seems that the text has not been checked. I suggest a review by a native speaker before a further submission.
Author Response
Referee 2
In this special issue of MDPI’s Brain Sciences on “Neurobiological Basis of Developmental Dyslexia” the guest editor invites to write a short review with the aim of select and summarize the most substantial contributions to the field that will help to come to a solid understanding of the etiopathological mechanisms at the basis of the disorder. Overall the information collected represents valuable evidence about the topic, but due to the non-homogeneous and non-systematic presentation of the contents (in some cases too detailed and in other too general, often repetitive and redundant), the manuscript becomes difficult and slow to read.
Author´s response: In my view, the assertion that the paper is non-homogenous and non-systematic, that some cases are too detailled and others are repetitive and redundant is unjustified. The reviewer does not specify which text is too detailled and where the text is redundant. Unnecessary repititions have been deleted; now no text is superfluous (except page 2 (Introduction), lines 13-20; paragraph 5, lines 11 and 12 which have been deleted). Paragraphs „Causation“, „Developmental dyslexia and hemianopic reading disturbance“, and „Differences between developmental dyslexia and simultanagnosia“ at the end of the introduction have been moved to „Supplemental Texts“.
The neurobiology of reading and DD cannot be understood without taking into account a detailled view on neuroanatomy. I described only the most important connections that are discussed in the literature, and I omitted recurrent connections to reduce the complexity for readers who are not familiar with neuroanatomy.
In my view, the text is clearly structured in: anatomy of the M- and P- cell pathways in the retina (paragraph 2), anatomy of the visual cortex (paragraph 3), the two visual systems hypotheses (paragraph 4), motion and contrast patterns in the dorsal and ventral stream (paragraph 5), the magnocell hypothesis of DD (paragraph 6), evidence that DD id due to a visual processing impairment (paragraph 7), the role oft he VWFA (paragraph 8) (structured in anatomy (8.1), studies about the reaction on words (8.2), pseudowords (8.3), and non-word stimuli (8.4), the hierachy of visual processing in the VWFA (8.5), the result of lesions to the VWFA (8.6), and neurobiological basis of graphem to phonem association (paragraph 9).
My first concern is about the length of the present work that might not fit the format requested, however, a detailed “Supplementary Information” section might be useful and could help to keep the manuscript in a short format.
Author´s response: I have been asked to write a contribution about neurobiological foundations of DD and not to write a short review. Reading is a complex performance for which several abilities are necessary and which involves the function of the occipital lobe temporal lobe parietal lobe and frontal lobe.The necessary anatomical and physiological basis is complex and must be described in a paper on the neurobiological basis of DD. Similarly, the studies on the response of brain structures to words, pseudowords, falsefonts, etc. must be described and cannot be neglegted. Similarly, reading disorders must be reported after damage to brain structures that have possible relevance to reading. These neuropsychological studies are the key to the understanding of neurobiological bases of reading.The literature is also controversial and the interpretation of the results are often questionable. A thorough presentation of the state of research and scientific discussion is not possible on a few pages.
Then, I have some concerns about the writing style and the organization of the sections. In particular, key critical points are:
- Aim of the review and Methods Section
The aim of the work is not extensively discussed, I suggest explaining the purpose of the author and the hypothesis at the basis.
Author´s response: The aim of the paper is now stated at the end of the Introduction: “The aim of the present paper is to examine whether research results supports the hypothesis that DD is caused by a phonological disorder, an impairemant of the magnocellular stream, or an impairment of the visual system and the visual word form area (VWFA). The paper also aimes to examine if the studies which claim to shed light on the causal relationship between reading performance and neurobiological processes satisfy the conditions that must be met to claim such a causal relationship.”
A Methods section is missing. It is necessary to clarify what kind of review it is: narrative, systematic, scoping, gap analysis…? Then the author should detail how the selection of the paper has been done and which criteria of eligibility he followed.
Author´s response: The following text has been added at the end of the Introduction: „To this end, several thousand papers about the anatomy, physiology, neuropsychology of the visual system in humans, about visual psychophysics and about dyslexia which were available in the Max- Planck-Institute for Psychiatry, the Bavarian State Library, the Library of the Medical Faculty of the University of Munich, pubmed, science direct, psycnet or other internet publishers, and which the author collected over a period of about 40 years up to the year 2021, were checked to decide if they were relevant to the questions posed in the present paper, 289 of the most relevant papers have been referenced.“
- Structure
The author should provide a brief summary at the end of each paragraph, which allows the reader to easily understand the results concerning the topic treated in that paragraph. The results are reported as a list, an overview is lacking.
Author´s response: The anatomy chapter has already been summarizes in Figure 1. I provided a summary at the end of paragraph 5:
„In brief it can be said that motion selective cells were found in area V1, V2, V3, V3A, MT/V5, V3A, V2, VP . Contrast selective cells in humans were present in areas V1 to V4 of the ventral stream, and in V3 and V3A and the intraparietal sulcus in the dorsal stream. Cells in areas V1, V2, V3, V3A and Vh4 preferred high temporal frequency stimuli.“
at the end of paragraph 6:
„In conclusion, the following can be assumed: It appears that a functional impairment of magnocells and parvocells of the visual system causes DD. There is, however, no sufficient evidence for the assumption that DD is only or predominantly caused by a magnocell deficit.
Studies which show that dyslexic children exert appropriate eye movements when their eye movements are guided by a computer, and that they can learn to exert appropriate eye movements within 30 minutes, demonstrate that children with DD don´t suffer from a lack of eye movement control due to a magnocell deficit.“
at the end of paragrapf 7:
„Taken together, experimental results [57-59] have hitherto shown that the ability to recognize several letters in a word or pseudoword simultaneously, essentially depends on the fixation time. Temporal summation [152-165] can explain the ability to recognize more letters in pseudowords simultaneously when the fixation time is prolonged. Regardless of the fixation time, there is an individual limit on how many letters a person can recognize simultaneously. The question arises whether dyslexia is only a consequence of inpaired visual cortex processing or whether brain structures that receive their input from the visual cortex are crucial. This brought a region in the middle fusiform gyrus into the focus of research. The question arises whether dyslexia is only a consequence of inpaired visual cortex processing or whether brain structures that receive their input from the visual cortex are crucial. This brought a region in the middle fusiform gyrus into the focus of research.“
and at the end of paragraph 8.5. (see below)
A table would be useful in order to organize all the study results cited in the review, especially the controversial ones. In each paragraph, the author should organize the scientific literature in a more systematic way, e.g.: dividing studies for population (clinical, general population); age (children, adult); methodology used (EEG, ERP, fMRI…).
Author´s response: In my view, the literature is clearly orgranized with respect to the questions „does the VWFA respond to real words“, does the VWFA respond to pseudowords, pseudohomophones, foreign writing and false fonds, „ does the VWFA respond to line drawings and pictures“. To that end the literature is clearly devided into studies which register the response of the VWFA to real words, pseudowords, pseudohomophones, foreign writing and false fonds, line drawings and pictures, and studies demonstrating which cerebral lesions result in which reading impairments. Almost all these studies are functional MRI studies. Only two studies are EEG studies because EEG recordings don´t have a sufficiently high spatial resolution. All MRI and EEG studies are methodologically in the same vein as they do not allow a conclusion on whether the function of the VWFA is a necessary or sufficient condition for reading. Lesion studies are summarized in a separate paragraph. They allow a conclusion on whether the function oft he VWFA is necessaty or sufficient for reading. Some VEP studies studies relate to visual cortex function to explain temporal summation (e.g. May et al. 1991; Lehmkuhle et al. 1993; Kubova et al. 2015), other EEG studies (e. g. Lochy et al 218) investigate the gyrus fusiformis. PET studies (e.G. Petersen et al 1988) and some MRI studies relate to visual cortex function many others to the reaction of the VWFA to words, pseudowords, pictures etc. These studies are divided according to their contribution to the answer of neurobiological questions. These studies should not be mixed by deviding them according to clinical, (general population); age (children, adult); methodology used (EEG, ERP, fMRI). In my view, deviding the studies further in clinical, (general population); age (children, adult); methodology used (EEG, ERP, fMRI) makes the presentation of the studies confusing and does not strengthen any argument. I prefer the reviewer´s suggestion to add a short summary after each paragraph. I repeated the numbers of the references in the summaries.
- Writing style
Many paragraphs (e.g.: 8.2, 8.3 and 8.4) should be summarized in order to assert clear conclusions (e.g.: Which kind of stimuli does the VWFA process?).
Author´s response:The following summary was added after Paragraph 8.5:
„Taken together, the result of the studies on the function of VWFA is that their neural networks are activated by real words, psedowords, pseudohomophones, strings of foreign letters (Amahri, Chinese, Korean, Hebrew) unknown to the reader, false fonts, line drawings and pictures [213-222]. These results demonstrate that the VWFA receives visual input but it does not demonstrate whether or not the function of the VWFA is a necessary or sufficient condition for normal reading. This can only be demonstrated by the results of damage to the VWFA.“
A short summary has been added at the end of paragraph 8.6:
„In summary, the finding that patients have difficulty reading pseudowords after damage to the VWFA suggests that the VWFA may have a role in simultaneous recognition of a string of letters [187, 251, 254, 255, 257, 258, 259]. Real words can be recognized better because they can be deduced when only a few letters and the shape of the word are seen at a time.“
I also suggest citing more recent and relevant literature, it seems that the author comes to conclusions based mainly on his own studies. Providing better citations will help place the current work in better context of the scientific progress in this area over the last years.
Author´s response: The assertion that I did not cite recent literatute is incorrect. 43 references are from the last 5 years and 20 references are from the last 3 years. With regard to the numerous publications it is of course not possible to include all published literature. I have selected the references that I consider most significant.
The conclusions are mainly based on my own studies because no other studies were based on a clear concept of cause. The research about the concept of causation has been ignored in all studies on dyslexia. Assertions about causal relations were made on a vague every day understanding but not on scientific grounds. Publications on „causation“ is quite mathematical and difficult to understand for readers who are not familiar with mathematical logics and set theory. I therefore presented a non mathematical criterion that is easy to understand.
I added the following text to clarify the logical requirements that must be met in order to make assertions about causation on paragraph 7, lines 5-11: „To prove a causal relationship between a stimulus parameter and reading performance it must be demonstrated that the realization of this stimulus parameter improves reading performance in a repeatable way and that the absence of this parameter worsens reading performance. The studies by Werth (2018, 2019, 2021) are the only studies in which the causal relationship has been examined in this way in a repeatable manner. To this end, pronounceable pseudowords of a length between 3 and 6 letters were displayed only once between 250 and 500 ms.“
There is also a methodological problem with most psychological studies: They usually don´t fulfill the requirements of the Americal Statisical Association (Gigerenzer 2004; Joannidis 2005; Wasserstein 2016; Wasserstein & Lazar 2016; Benjamin et al. 2018; Billheimer 2019; Wasserstein et al. 2019). The many labs study has shown that most psychological results – studies on dyslexia are no exception- are not reproducible (Klein et al. 2018). It is required that studies are repeated and do not only rest on p-value significance statistics but on effect size statistics (Cohen, Hedges etc.). The studies by Werth (2018, 2019, 2021) belong to the minority of studies that fulfill these requirements. Therefore, one can be shure that the results are substanciated.
In the text there are several typographical error or misprint (e.g.: lines 57,81,182,237, 451) and inconsistent formatting (e.g.: lines 100-102; 422-431; 731-732; 761-762), it seems that the text has not been checked. I suggest a review by a native speaker before a further submission.
Author´s response: The paper has been checked by a native speaker and all typographical errors have been corrected.